# Different TCR-induced T lymphocyte responses are potentiated by stiffness with variable sensitivity

Michael Saitakis[1], Stéphanie Dogniaux[1], Christel Goudot[1], Nathalie Bufi[2], Sophie Asnacios[2,3], Mathieu Maurin[1], Clotilde Randriamampita[4], Atef Asnacios[2]*, Claire Hivroz[1]*

[1]Institut Curie Section Recherche, INSERM U932 & PSL Research University, Paris, France; [2]Laboratoire Matières et systèmes complexes, Université Paris-Diderot and CNRS, UMR 7057, Sorbonne Paris Cité, Paris, France; [3]Department of Physics, Sorbonne Universités, UPMC Université Paris, Paris, France; [4]INSERM, U1016, Institut Cochin & UMR8104, CNRS & Université Paris Descartes, Sorbonne Paris Cité, Paris, France

**Abstract** T cells are mechanosensitive but the effect of stiffness on their functions is still debated. We characterize herein how human primary CD4$^+$ T cell functions are affected by stiffness within the physiological Young's modulus range of 0.5 kPa to 100 kPa. Stiffness modulates T lymphocyte migration and morphological changes induced by TCR/CD3 triggering. Stiffness also increases TCR-induced immune system, metabolism and cell-cycle-related genes. Yet, upon TCR/CD3 stimulation, while cytokine production increases within a wide range of stiffness, from hundreds of Pa to hundreds of kPa, T cell metabolic properties and cell cycle progression are only increased by the highest stiffness tested (100 kPa). Finally, mechanical properties of adherent antigen-presenting cells modulate cytokine production by T cells. Together, these results reveal that T cells discriminate between the wide range of stiffness values found in the body and adapt their responses accordingly.

*For correspondence: atef. asnacios@univ-paris-diderot.fr (AA); claire.hivroz@curie.fr (CH)

**Competing interests:** The authors declare that no competing interests exist.

## Introduction

During their life time, T cells are exposed to a wide range of biochemical and mechanical environments, which range from soft tissues like thymus or bone marrow, to stiffer peripheral inflamed tissues. Moreover, in order to mount their specific immune response, CD4$^+$ T cells need to interact and form immune synapses with a wide range of antigen-presenting cells (APCs), which are exposed to a variety of stimuli (*Thauland and Parker, 2010*) and display a range of stiffness values (*Bufi et al., 2015*). While the biochemistry of such interactions has been extensively explored, the effect of the mechanical landscape on T cell responses has only recently been addressed. Several studies have investigated the role of mechanics in T cells (*Chen and Zhu, 2013*): the T cell receptor (TCR) itself has been shown to act as a mechanosensor (*Kim et al., 2009*; *Lee et al., 2015*; *Li et al., 2010*); T cells were found to generate forces upon activation and costimulation (*Bashour et al., 2014*; *Hu and Butte, 2016*; *Husson et al., 2011*; *Ma et al., 2016*; *Liu et al., 2016*); T cells were also found to adapt to the stiffening of an artificial APC by changing the loading rate of their pulling forces (*Husson et al., 2011*).

In terms of mechanical parameters, stiffness is of particular interest. Cells and tissues display vastly different stiffness values compared to extensively used glass and plastic-ware. Human T cells are extremely soft, in the range of 0.1 kPa (*Bufi et al., 2015*; *Guillou et al., 2016*), skeletal muscle

**eLife digest** Our immune system contains many cells that play various roles in defending the body against infection, cancer and other threats. For example, T cells constantly patrol the body ready to detect and respond to dangers. They do so by gathering cues from their surroundings, which can be specific chemical signals or physical properties such as the stiffness of tissues. Once the T cells are active they respond in several different ways including releasing hormones and dividing to produce more T cells.

Tissue stiffness varies considerably between different organs. Furthermore, disease can lead to changes in tissue stiffness. For example, tissues become more rigid when they are inflamed. The stiffness and other physical properties of the surfaces that T cells interact with affect how the cells respond when they detect a threat, but few details are known about exactly how these cues tune T cell responses. Saitakis et al. studied how human T cells respond to artificial surfaces of varying stiffness that mimic the range found in the body.

The experiments show that T cells that interact with stiff surfaces become more active than T cells that interact with softer surfaces. However, some responses are more sensitive to the stiffness of the surface than others. For example, the ability of the T cells to release hormones was affected by the whole range of stiffnesses tested in the experiments, whereas only very stiff surfaces stimulated the T cells to divide.

These findings show that T cells can detect the stiffness of surfaces in the body and use this to adapt how they respond to threats. Future challenges will be to find out how T cells sense the physical properties of their surroundings and investigate whether cell and tissue stiffness affects immune responses in the body. This will help us to understand how T cells fight infections and other threats, and could be used to develop new ways of boosting these cells to fight cancer and other diseases.

stiffness is in the range of tens of kPa (*Uffmann et al., 2004*), while calcified bone can reach values of GPa (*Gasiorowski et al., 2013*). For many cell types, stiffness has been shown to influence cell growth, differentiation, migration and survival (*Pelham and Wang, 1997*; *Lo et al., 2000*; *Discher et al., 2005*; *Engler et al., 2006*; *Solon et al., 2007*; *Oakes et al., 2009*; *Trappmann et al., 2012*). In T cells, substrate stiffness was shown to affect T cell activation (*Judokusumo et al., 2012*; *O'Connor et al., 2012*; *Tabdanov et al., 2015*) and to regulate the forces exerted by T cells on their substrates (*Husson et al., 2011*; *Hui et al., 2015*). Moreover, it was also recently proposed to regulate the cytotoxicity of T lymphocytes (*Basu et al., 2016*). However, most studies, especially for CD4[+] T cells, were looking into a handful of parameters and/or used a stiffness range that did not take into account the stiffness of myeloid APCs (0.2 to 1.5 kPa), which was shown to vary according to APC type and inflammatory conditions (*Bufi et al., 2015*).

We addressed herein the impact of substrate stiffness on different key functions of T lymphocytes using a physiologically relevant rigidity range (from hundreds of Pa to 100 kPa). Our results show that migration, gene expression, cytokine secretion, metabolism and cell cycle progression of human CD4[+] T lymphocytes are affected by stiffness of the activating substrates. They also reveal that some functions are sensitive to this potentiation from a low rigidity range (in the order of magnitude reported for APC), while others are mostly affected by higher rigidity (in the order of magnitude reported for tissues). Therefore, human CD4[+] T cells can modulate their responses according to their mechanical environment, with the TCR being an exquisite mechanosensor.

## Results

### T cell migratory properties and morphology are modulated by substrate stiffness

Effector T cells have to migrate outside the lymph nodes and in the periphery (*Weninger et al., 2001*; *Henrickson et al., 2008*), where they can come across a wide variety of cell and tissue stiffness (*Mathur et al., 2001*; *Kataoka et al., 2002*; *Bufi et al., 2015*). Whereas the elastic properties

of substrates have been shown to modulate the migratory capacities of many cell types (*Lo et al., 2000*; *Zaman et al., 2006*; *Ghosh et al., 2007*; *Oakes et al., 2009*), not much is known on the effect of these properties on T lymphocyte migration. We thus studied the migration of human CD4[+] T lymphoblasts on poly-acrylamide gels (PA-gels) of varying stiffness values. PA-gels contained streptavidin-acrylamide molecules that allow the specific immobilization of biotinylated molecules. Their elastic moduli were measured with a parallel-plate shear rheometer (*Figure 1—figure supplement 1A*). We used PA-gels with three different Young's modulus values (which is a measure of substrate stiffness): 0.5, 6.4 and 100 kPa (*Table 1*) that respectively mimic the typical stiffness range reported for human monocyte-derived dendritic cells (*Bufi et al., 2015*), human endothelial cells (*Mathur et al., 2001*; *Kataoka et al., 2002*) and tendon tissue or bone (*Oakes et al., 2009*; *Gasiorowski et al., 2013*). PA-gels, which do not allow for non-specific adsorption of serum proteins (*Figure 1—figure supplement 1B*), were coated with recombinant biotinylated ICAM-1 (intercellular adhesion molecule-1)/Fc chimeric molecules. ICAM-1, an adhesion molecule present on the surface of activated APCs and endothelial cells, is the ligand of the integrin LFA-1 and a key regulator of T cell migration and activation (*Altmann et al., 1989*; *Van Seventer et al., 1990*). Following ICAM-1 binding, LFA-1 can be modulated by mechanical forces (*Springer and Dustin, 2012*; *Chen and Zhu, 2013*) and promote T cell migration (*Morin et al., 2008*; *Jacobelli et al., 2009*). To induce TCR/CD3 stimulation, biotinylated anti-CD3ε antibodies (hereafter referred to as aCD3) were added together with biotinylated anti-CD28 (hereafter referred to as aCD28) and biotinylated ICAM-1/Fc on the PA-gels. The overall protein coating of PA-gels was similar on the different PA-gels (*Figure 1—figure supplement 1C*). The surface density of aCD3 was calculated to be in the range of 2 to 4 molecules/ $\mu m^2$, that is in the low range of agonist densities used in other studies (*Grakoui et al., 1999*; *Krummel et al., 2000*; *Purtic et al., 2005*; *Varma et al., 2006*). It roughly corresponded to 100 to 1000 MHC-peptides per APC, close to the minimum number of pMHC complexes reported to induce cytokine production by T cells (*Harding and Unanue, 1990*; *Huang et al., 2013*). In these coating conditions, the molar ratio of ICAM-1 to aCD3 was 10 fold, therefore ICAM-1-Fc density was calculated to be in the range of 20 to 40 molecules/$\mu m^2$.

Migration was monitored in the first 20–30 min of contact by live microscopy on PA-gels coated with biotinylated ICAM-1 alone (*Videos 1–3*) or together with aCD3+aCD28 antibodies (*Videos 4–6*). Tracks of individual cells were obtained (*Figure 1—figure supplement 1D*) along with mean instantaneous velocities and maximum distance travelled on the gels. We considered cells as either arrested, if the maximum distance they travelled in 5 min was lower than 10 $\mu$m, or migrating, if their maximum distance was higher. For ICAM-1 coated PA-gels, the mean instantaneous velocity of migrating T cells (*Figure 1A*) was significantly but modestly higher for the stiff 100 kPa gel (15.5 ± 0.4 $\mu$m/min) than for the 0.5 kPa (12.4 ± 0.3 $\mu$m/min) and 6.4 kPa gel (13.1 ± 0.3 $\mu$m/min). Similarly, calculating the maximum distance covered by the T cells on the substrate after 5 min revealed that T cells migrated over longer distances on the 100 kPa gels than on the 0.5 kPa and 6.4 kPa gels (*Figure 1B*).

TCR/CD3 triggering has been shown to induce a stop signal, that is, an arrest in T cell migration (*Dustin et al., 1997*). We thus investigated whether the rigidity of the surface bearing the TCR ligand can affect this stop signal. The number of arrested cells was measured on the PA-gels of varying stiffness. The percentage of arrested T cells in the presence of aCD3+aCD28 was higher on the stiff 100 kPa gels than on the softer ones (*Figure 1C*). Yet, percentage of arrested cells never reached the level obtained on glass coated with aCD3+aCD28+ICAM-1. Moreover, aCD3 coupled

**Table 1.** Composition of PA-gels and equivalent elastic modulus values (mean values with standard error).

| Acrylamide (% w/v) | bis-Acrylamide (% w/v) | Young's modulus $E$ (Pa) |
| --- | --- | --- |
| 3 | 0.2 | 513 ± 48 |
| 5 | 0.5 | 6416 ± 228 |
| 18 | 0.38 | 100,000 * |

*(reported in *Trappmann et al., 2012*)



**Video 1.** Live microscopy video of T lymphoblasts on a PA-gel of 0.5 kPa coated with ICAM-1-Fc.

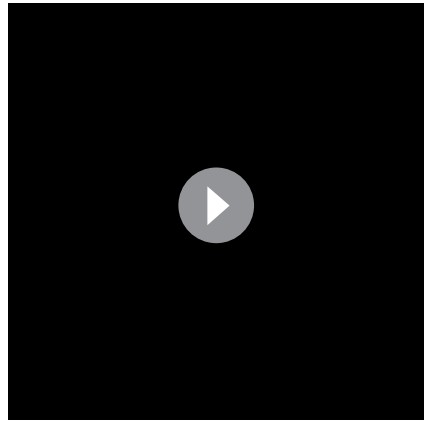

**Video 2.** Live microscopy video of T lymphoblasts on a PA-gel of 6.4 kPa coated with ICAM-1-Fc.

to 0.5 and 6.4 kPa did not induce any significant deceleration of lymphoblasts, while it did induce deceleration of non-arrested lymphoblasts when coupled to 100 kPa gels (*Figure 1—figure supplement 1E*).

To better characterize interaction of T cells with the PA-gel-activating substrates, we studied the morphology of T lymphoblasts by scanning electron microscopy. T cells characteristically spread within minutes on substrates coated with TCR-activating molecules (antibodies and/or pMHCs) and adopt a flat symmetrical round shape (*Bunnell et al., 2001*; *Brodovitch et al., 2013*). This was mostly observed for commonly used substrates such as glass or plastic, which have stiffness values in the order of $10^7$ kPa (or tens of GPa), and for glass-supported lipid bilayers. Yet, the morphology adopted by T cells on substrates of physiological stiffness has scarcely been addressed. T cells were activated for 30 min on PA-gels of varying stiffness and on glass coverslips coated with aCD3+-aCD28+ICAM-1. Because of the multiple treatments of the slides and washing conditions, we mostly imaged T lymphoblasts interacting strongly with the substrates (arrested cells) and very few migrating cells were observed. Softer gels (0.5 and 6.4 kPa) allowed for minimal spreading of some T cell extensions, whereas the T cell body did not spread. Contact with the stiffest gel (100 kPa) induced more pronounced spreading of cell extensions (*Figure 1D*). Thus, the cell-surface contacts seemed to demonstrate distinct phases of T cell spreading. Protrusions on 0.5 and 6.4 kPa gels were thin (100–200 nm). On 100 kPa gels, protrusions were similarly thin until they reached the surface, where they spread. These structures were reminiscent of the invadosome/podosome-like protrusions (ILPs) reported during T cell scanning of activated endothelial cells that have rigidities in the order of kPa (*Sage et al., 2012*; *Kumari et al., 2015*). On glass, T cells were extensively spread, with all the ventral membrane forming a single thin lamellipodium, as previously shown (*Bunnell et al., 2001*). Yet, these last results could not be only interpreted in terms of substrate stiffness, since, first, antibody coating on glass was twice as high as on PA-gels and, second, non-specific adsorption of proteins from serum and/or from cells could occur on glass, whereas this phenomenon was minimal on PA-gels (*Figure 1—figure supplement 1B and C*). However, it is worth noting that the 'fried egg' shape, reported in most reviews as the classical morphology of a T cell forming a mature synapse, was neither observed when T cells interacted with substrates of physiological stiffness



**Video 3.** Live microscopy video of T lymphoblasts on a PA-gel of 100 kPa coated with ICAM-1-Fc.

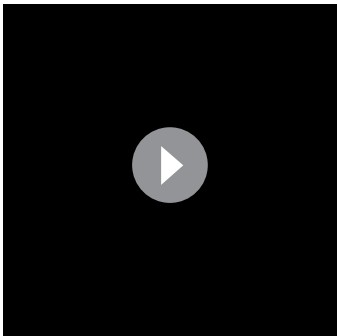

**Video 4.** Live microscopy video of T lymphoblasts on a PA-gel of 0.5 kPa coated with aCD3+aCD28+ICAM-1-Fc. DOI: 10.7554/eLife.23190.009

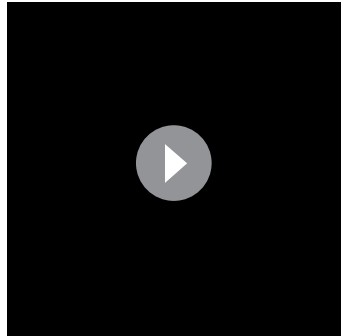

**Video 5.** Live microscopy video of T lymphoblasts on a PA-gel of 6.4 kPa coated with aCD3+aCD28+ICAM-1-Fc. DOI: 10.7554/eLife.23190.010

range (herein) nor when in contact with dendritic cells (*Trautmann and Valitutti, 2003*). Such differences were also reported for primary fibroblasts (*Gu et al., 2014*) that formed robust focal adhesions and underwent directed migration on stiff substrates but switched to ILP formation and invasive behavior on substrates of low stiffness (in the order of few hundreds of Pa). Thus, within a physiological stiffness range, T cells scan their substrate through thin protrusions but do not spread extensively their cell body in response to TCR activation

These data provide meaningful information on T cell behavior on elastic substrates of physiological stiffness. The higher velocity of T cells on stiffer ICAM-1 coated PA-gels is at odds with results reported for fibroblasts (*Lo et al., 2000*; *Ghosh et al., 2007*), smooth muscle cells (*Zaman et al., 2006*) and neutrophils (*Oakes et al., 2009*), which show an inverse correlation between stiffness of substrate and velocity. This might be attributed to differences not only in the cell types used but also in the integrin ligand used (collagen or fibronectin versus ICAM-1 in our case). Moreover, our results suggest that at low density of TCR ligands and in the range of physiological APC rigidities (*Bufi et al., 2015*), T lymphocytes do not stop to form a synapse but rather form kinapses (*Dustin, 2008*). Yet, they arrest more in more acute conditions of stiffness (hundreds of kPa).

## Gene expression of CD4[+] T cells shows a graded response to stiffness

To gain more insight into the stiffness-mediated modulation of T cell activation, we performed a RNA microarray analysis of effector CD4[+] T cells activated for 24 hr on PA-gels of varying stiffness coated with either aCD28+ICAM-1 alone or together with aCD3. Principal component analysis (PCA) was performed on the full data set of the three different stiffness values (0.5, 6.4, and 100 kPa) for these two coating conditions (*Figure 2A*). This analysis revealed that the main differences in gene expression were observed between the conditions with/without aCD3. Moreover, the effect of substrate stiffness on T cell gene expression was mainly observed when the TCR/CD3 was stimulated.

We then performed differential analysis on the genes that showed significantly changed expression when comparing presence and absence of aCD3 stimulation for each stiffness value. The number of genes with significantly changed expression was very low for T cells on the soft 0.5 kPa gel (34 up-regulated and seven down-regulated) and increased on the stiffer 6.4 kPa (2007 up and 1011 down) and 100 kPa gels (2611 up and 1744 down) (*Figure 2B*, *Supplementary file 1*). This suggests that the TCR/CD3 is herein the main rigidity sensing receptor and that it induces stronger signaling when its ligand is attached to stiff rather than soft substrates.

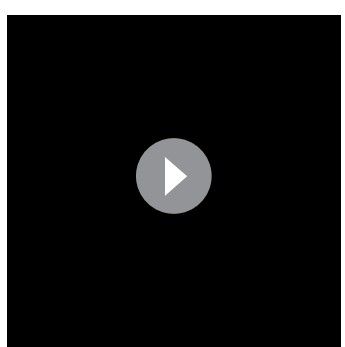

**Video 6.** Live microscopy video of T lymphoblasts on a PA-gel of 100 kPa coated with aCD3+aCD28+ICAM-1-Fc. DOI: 10.7554/eLife.23190.011

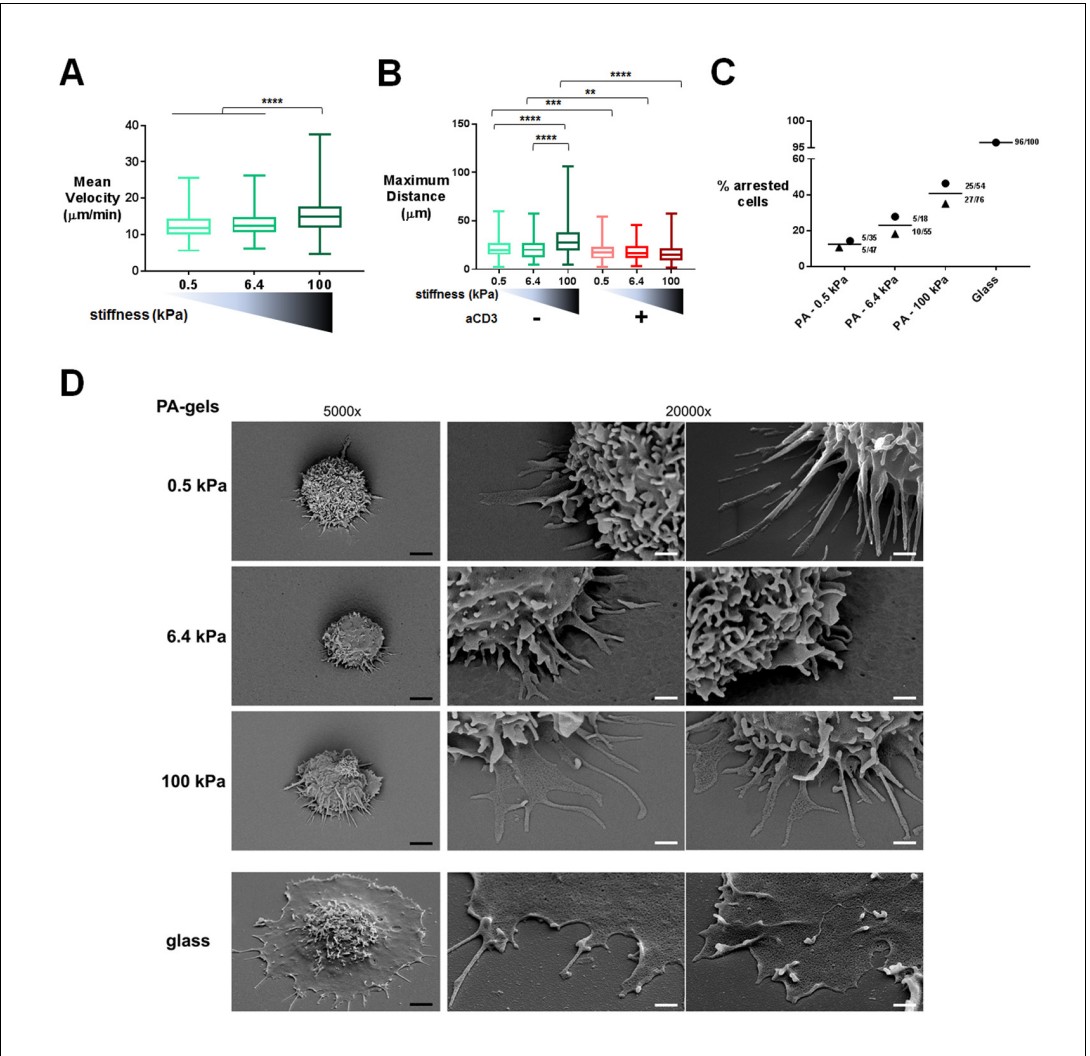

**Figure 1.** T cell migratory properties and morphology are modulated by substrate stiffness. (**A**) Mean instantaneous velocities of migrating T cells on ICAM-1-coated PA-gels of varying stiffness (n_cells: 50–100 for each condition from n_Donors: 4). (**B**) Maximum distance travelled by T cells on PA-gels of varying stiffness for a duration of 5 min (n_cells: 50–100 for each condition from n_Donors: 4). Boxes and whiskers for minimum and maximum are shown. For statistical analysis, unpaired parametric t-tests were performed: ****p-value<0.0001, ***p-value<0.001, **p-value<0.01. (**C**) Percentage of arrested cells on aCD3+aCD28+ICAM-1 coated PA-gels of varying stiffness. T cell response on glass coated with aCD3+aCD28+ICAM-1 is shown for comparison. Mean values and number of cells per condition are shown (n_Donors: 2). (**D**) Scanning electron microscopy pictures of T cells (representative of n_cells: 5 per condition from n_Donors: 2) on aCD3+aCD28+ICAM-1-coated substrates for two magnifications (5000x and 20000x). Black scale bars: 2 μm, white scale bars: 500 nm.
The following figure supplement is available for figure 1:

**Figure supplement 1.** Characterization of PA-gels and additionnal data on migration.

---

The expression of several genes specific of T cell response was analyzed. Cytokines (TNF, IFNG, LTA, IL22, IL17A), T cell surface markers (BTLA, IL4R, CD40LG, IL2RA, CD69, ICOS), T-cell-specific transcription factors (TBX21 and FOXP3) and the proliferation transcription factor MYC all showed enhanced gene expression with increasing stiffness in the presence of aCD3 (*Figure 2C*). Gene expression of the translation initiation factor EIF4E, which is up-regulated with T cell activation (*Bjur et al., 2013*), and the metabolic regulator for promoting glycolysis HIF1A (*Pollizzi and Powell, 2014*) was also increased in response to stiffness. Of note, expression of lamin genes (LMNA,

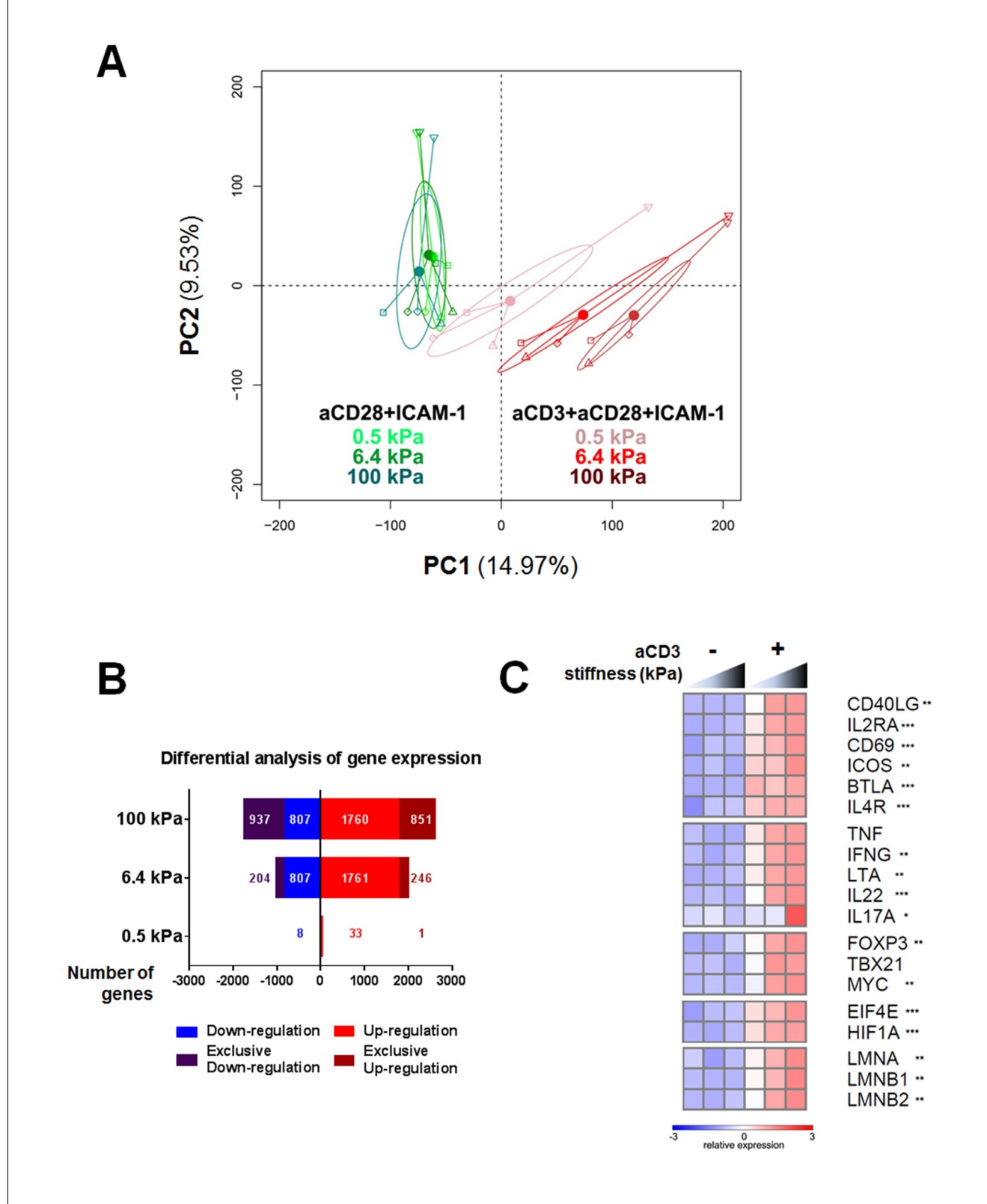

**Figure 2.** Gene expression of CD4[+] T cells shows a graded response to stiffness. (**A**) Principal component analysis reveals that gene expression is modulated by T cell substrate stiffness only in presence of aCD3 ($n_{Donors}$: 4). (**B**) Number of genes that displayed differential expression between the conditions with and without aCD3 on PA-gels of varying stiffness. 'Exclusive' indicates the genes that are found Up- or Down-regulated only at a given

*Figure 2 continued on next page*

*Figure 2 continued*

stiffness value. (C) Relative expression of T cell related genes following Affymetrix microarray analysis. Asterisks indicate the presence of these genes in the differential analysis: *** for 0.5 to 100 kPa, ** for 6.4 to 100 kPa, * for 100 kPa only.

LMNB1, LMNB2), which has been shown for other cell types to scale with tissue stiffness (*Swift et al., 2013*), was gradually increased with stiffness on aCD3-coated PA-gels (*Figure 2C*).

Overall, these results show that TCR induced gene expression is potentiated by the stiffness of the surface presenting TCR ligands and genes related to CD4$^+$ T cell immune response are particularly sensitive.

## Stiffness potentiates TCR/CD3-induced transcriptional response

In a founding study (*Engler et al., 2006*), substrate stiffness was shown to induce different programs of differentiation of mesenchymal stem cells. We thus asked whether different stiffness values could induce different programs of T cell differentiation or whether stiffness acts instead as a rheostat on TCR/CD3-induced activation.

We first used the microarray data to perform pair-wise comparisons (presence vs. absence of aCD3 for each stiffness value) with the gene set enrichment analysis (GSEA) method. We used the publicly available Gene Ontology – Biological Processes (GO-BP) and the Kyoto Encyclopedia of Genes and Genomes (KEGG) pathway gene sets to identify the relatively enriched collections of genes and then extracted the enriched gene sets for each condition (*Figure 3A*, *Supplementary file 2*). Results showed that most of the enriched gene sets were common for the three different substrates and these included cell-cycle-related and immune-response-related processes. The stiff 100 kPa gel caused the enrichment of a higher number of gene sets in response to aCD3 stimulation, due to specific enrichment of gene sets such as mitochondrial biogenesis, oxidative phosphorylation and glycolysis (*Supplementary file 2*).

Pathway analysis on the differentially up-regulated genes, using the GO and the KEGG databases, showed that the 6.4 and 100 kPa PA-gels induced mostly common pathways, which were mitotic cell cycle, transcription and translation related (*Figure 3B*, *Figure 3—figure supplement 1*, *Supplementary files 3*, *4*). For the 0.5 kPa PA-gel, the two most prominent pathways induced were T cell activation and differentiation (*Figure 3B*), showing that the TCR was responsive from the softest end of the stiffness range. The two most prominent pathways induced *only* on 6.4 kPa gels included survival and apoptotic signaling pathways (*Supplementary file 3*) despite the fact that T cell viability was the same for all conditions (data not shown). Moreover, this was due to the differential expression of very few genes (e.g. CUL2, SART1, BRCC3 and ERCC8). The two most prominent pathways induced *only* on the 100 kPa gels included respiratory electron transport (e.g. genes of ATP synthase) and translation-related pathways (ribosomal protein genes) (*Figure 3B*, *Supplementary files 3*, *4*), showing that induction of these pathways required the stronger signaling provided by TCR ligands on the stiff substrate.

Finally, we performed K-means hierarchical clustering to look into particular patterns of gene expression upon TCR/CD3 activation in response to stiffness. Three major clusters were identified (*Figure 3C*): one with strong up-regulation (containing 1022 probes), one with weak down-regulation (containing 4412 probes) and one with weak or no up-regulation (containing 5928 probes). Pathway analysis of the strongly up-regulated cluster demonstrated enrichment of mostly cell cycle processes, cytokine signaling, nucleotide biosynthesis and ribosomal biogenesis (*Supplementary file 5*). The strongly up-regulated cluster showed a response to aCD3 that gradually increased with stiffness.

To get more insight into the sensitivity of up-regulated genes to stiffness, we used this strongly up-regulated gene cluster to look into changes in relative expression between the PA-gels of varying stiffness. Gene expression changes induced in T cells on 6.4 kPa gels relative to 0.5 kPa gels were plotted against gene expression changes induced on 100 kPa gels relative to 6.4 kPa gels (*Figure 3D*). The majority of the genes showed an expression that was more increased from 0.5 kPa to 6.4 kPa (right of the diagonal) than from 6.4 kPa to 100 kPa (left of the diagonal) (*Figure 3D*). These genes included cytokines and related genes: IL22, IFNG, TNF, LTA, CD40LG, IL2RA, IRF4; transcription factors: FOXP3, TBX21, MYC; and metabolism-related genes: LDHA, ATP8B4. Many genes showed expression that continued to rise for both transitions (genes close to the diagonal),

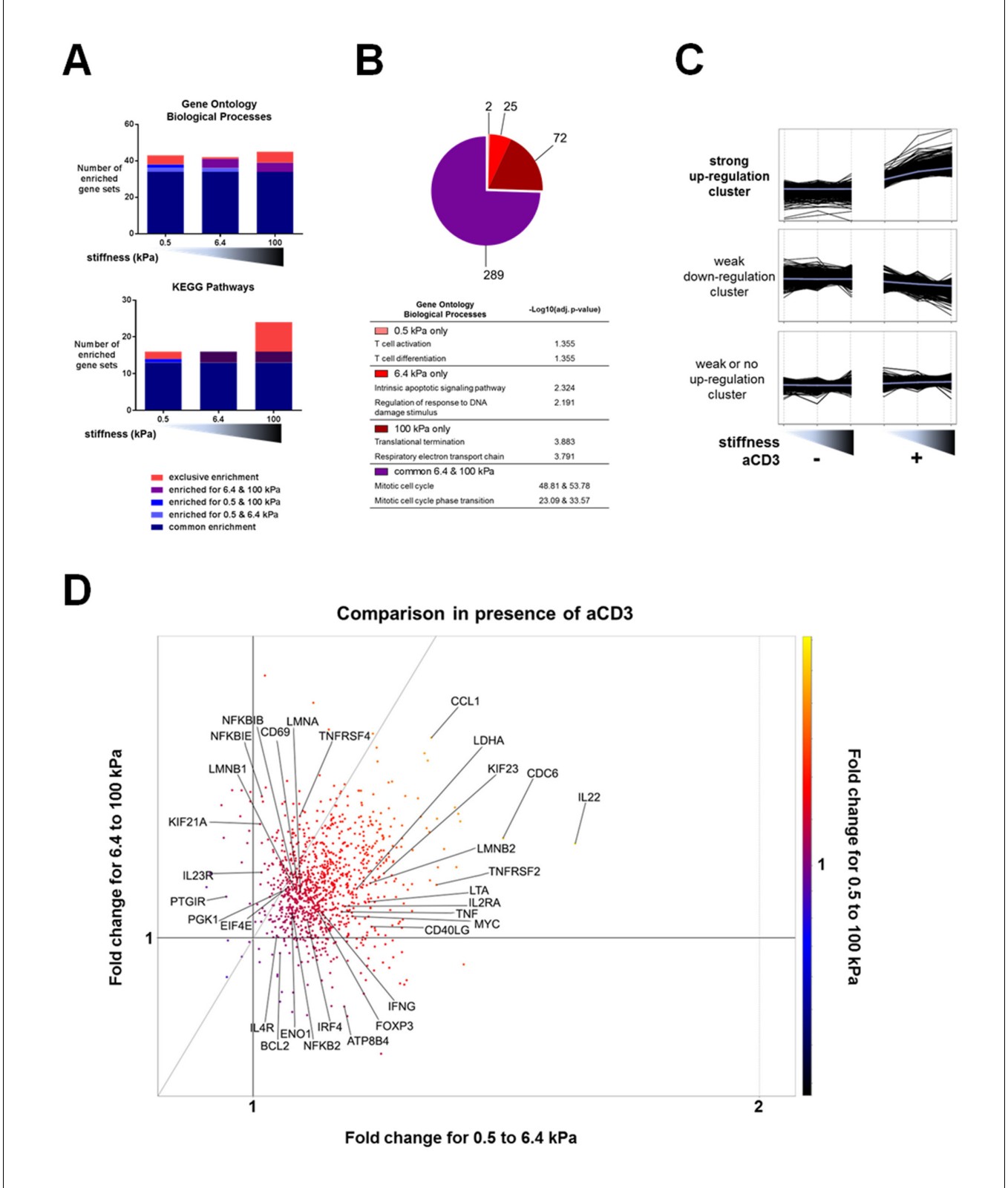

**Figure 3.** Stiffness potentiates TCR/CD3-induced transcriptional response. (A) Enrichment of gene sets on PA-gels of varying stiffness for p-values lower than 0.05, false discovery rates lower than 0.25 and NES values higher than 1.75. (B) Pathway analysis with the GO – BP database for differentially expressed genes between the conditions with and without aCD3 on PA gels of varying stiffness. The number of different pathways (pie-chart) and the top 2 hits of the enriched pathways, along with their negative log adjusted p-value (table), are shown. (C) K-means clustering of genes with different

*Figure 3 continued on next page*

*Figure 3 continued*

expression profiles on PA-gels of varying stiffness demonstrates three different clusters: one with strong up-regulation in the presence of aCD3 (containing 1022 probes), one with weak down-regulation (containing 4412 probes) and one with weak or no up-regulation (containing 5928 probes). (D) Comparison of the relative changes in gene expression in the presence of aCD3 for the strong up-regulation cluster. The x-axis shows the difference in gene expression for the transition of 0.5 to 6.4 kPa, the y-axis for the transition of 6.4 to 100 kPa and the colour gradient for the transition of 0.5 to 100 kPa.

The following figure supplement is available for figure 3:

**Figure supplement 1.** Pathway analysis with the KEGG database for differentially expressed genes between the conditions with and without aCD3 on PA gels of varying stiffness.

such as IL4R, CD69, NFKBIB, ENO1, PGK1, LMNA, and LMNB1. Other less numerous genes showed increased expression for the 6.4 to 100 kPa transition (IL23R, KIF21A, UBE2M, TNFRSF4).

The results from these three analyses show that expression of genes involved in the main TCR/CD3 induced T cell functions is sensitive to stiffness, with some functions (cytokine signaling, T cell activation) being induced in the low range of stiffness values and others (respiratory electron transport and glycolysis) requiring higher stiffness to be induced.

## Cytokine production is sensitive to a wide range of stiffness

Our transcriptomic analysis revealed that TCR-induced cytokine gene expression is sensitive to the whole range of substrate stiffness tested, increasing from 0.5 kPa to 100 kPa. Since sensitivity of cytokine production to substrate stiffness is currently a matter of debate (*Judokusumo et al., 2012*; *O'Connor et al., 2012*; *Hui et al., 2015*; *Tabdanov et al., 2015*; *Hivroz and Saitakis, 2016*), we went on to measure actual cytokine production. Human CD4$^+$ T lymphoblasts were cultured on PA-gels in the same conditions as for the transcriptomic analysis and IFNγ and TFNα production was measured by ELISA in the supernatants of 24 hr cultures. CD25, a CD4$^+$ T cell activation marker and protein product of IL2RA gene, was also measured by flow cytometry on the cell surface of T lymphoblasts. Cytokine production showed a graded response to stiffness in the presence of aCD3 (*Figure 4A and B*) confirming the results obtained on gene expression. A similar potentiating effect of stiffness was observed when ten times lower amount of aCD3 was used (*Figure 4—figure supplement 1A*). In contrast, CD25 expression, which was already present on the surface of T lymphoblasts, was only increased when T cells were activated on the aCD3+aCD28+ICAM-1-coated stiff 100 kPa gels (*Figure 4C and D*), whereas its gene expression was increased from 0.5 kPa (*Figure 2C*). These last results may be explained by the differences between transcriptomic and proteomic analyses that have already been reported (*Hukelmann et al., 2016*). They might also reflect a difference in sensitivity for the different measurements performed.

We then sought to confirm whether stiffness sensing by T cells required physical interaction of the TCR/CD3 complex to the gel substrate. CD4$^+$ T lymphoblasts were cultured on PA-gels of varying stiffness coated with ICAM-1 and soluble aCD3+aCD28 were added. In these conditions, IFNγ production was not affected by gel stiffness (*Figure 4—figure supplement 1B*) showing that mechanosensing indeed requires direct engagement of the TCR/CD3 complex with ligands on substrates of varying stiffness. Moreover, in the absence of ICAM-1 (PA-gels coated solely with biotinylated aCD3+aCD28), production of cytokines still followed substrate stiffness (*Figure 4—figure supplement 1C*) showing that ICAM-1 is not essential for the stiffness-mediated modulation of cytokine production.

These results were confirmed on freshly isolated memory CD4$^+$ T cells, which also showed increased production of cytokines (IFNγ and IL-2) and increased expression of the activation marker CD69 in response to increasing stiffness (*Figure 4—figure supplement 1D*).

Overall, our results show that TCR/CD3-induced cytokine production by effector and memory human CD4$^+$ T cells is increased by the rigidity of substrates bearing TCR ligands and shows exquisite sensitivity within the physiological range of cellular (0.5 and 6.4 kPa) and tissue rigidity (100 kPa).

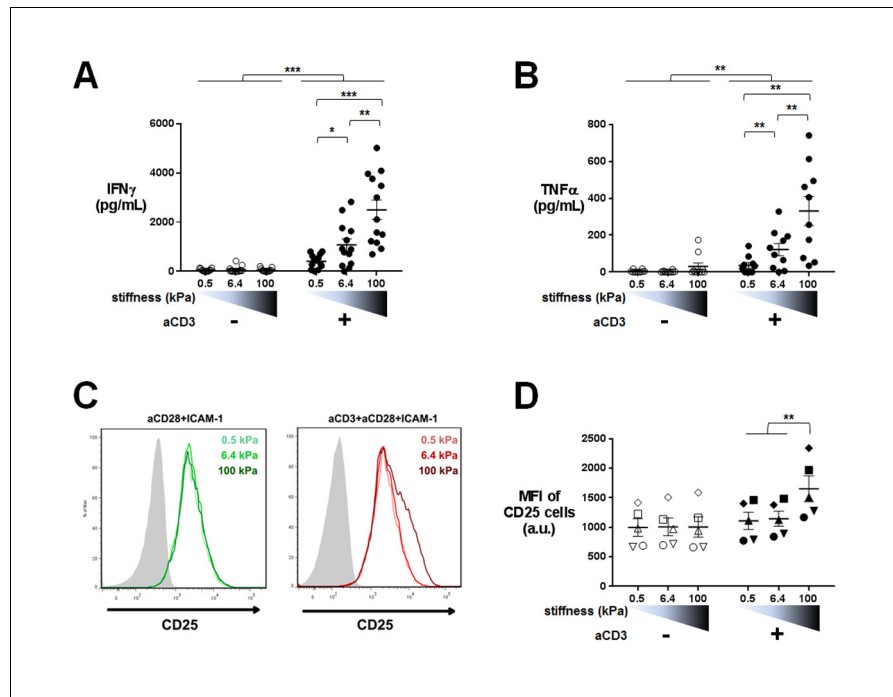

**Figure 4.** Cytokine production is sensitive to a wide range of stiffness. Production of (**A**) IFNγ (n$_{Donors}$: 13) and (**B**) TNFα (n$_{Donors}$: 10) on PA-gels of varying stiffness. In the presence of aCD3, the aCD3:aCD28 coating ratio was 1:10. (**C**) FACS plot of CD25 staining. A representative experiment is shown. (**D**) Mean fluorescence intensity of CD25-stained cells (n$_{Donors}$: 5). Mean values with standard error are shown. For statistical analysis, paired parametric t-tests were performed: ***p-value<0.001, **p-value<0.01, *p-value<0.05.

The following figure supplement is available for figure 4:

**Figure supplement 1.** Cytokine production on PA-gels: additionnal data.

## TCR-induced metabolic switch, cell cycle progression and proliferation are potentiated by stiffness

Gene expression profiles suggested that T cell metabolism was sensitive to the stiffness of PA-gels. This was in line with the fact that T cells dramatically change their metabolic activity, switching from a metabolically quiescent state to aerobic glycolysis upon TCR triggering and co-stimulatory activation (*Pearce et al., 2013*). To confirm T cell metabolic response to stiffness, we first measured the production of lactate in supernatants of cultures on PA-gels. The presence of aCD3 on PA-gels induced more lactate production by T cells than aCD28+ICAM1 alone (*Figure 5A*). This increase was measurable 16 hr after activation but was only significant for T cells activated on 100 kPa. After 24 hr, lactate production was significantly higher in the presence of aCD3, similar when cells were activated on 0.5 and 6.4 kPa PA-gels and more pronounced on the stiff 100 kPa PA-gel (*Figure 5A*). These results show that the T cell glycolytic switch induced by TCR triggering was increased by high stiffness values (100 kPa) but not by low ones (hundreds to thousands of Pa).

Several signaling pathways support and regulate the change in T cell metabolic program after activation (*O'Neill et al., 2016*; *Dimeloe et al., 2017*). A key metabolic regulator is the kinase mammalian target of rapamycin (mTOR), which promotes aerobic glycolysis and supports T cell growth and function (*Buck et al., 2015*). One of its regulatory functions is the increase in protein translation, via its canonical target p70 S6 kinase 1 (S6K1), which in turn phosphorylates p70 S6 ribosomal protein (rpS6) (*Laplante and Sabatini, 2012*). We monitored the activation of this signaling pathway by following the phosphorylation of rpS6 (Ser235/236) in T lymphoblasts activated as before on PA-gels. In the presence of aCD3, the percentage of phospho-rpS6+ T cells started to increase from 6

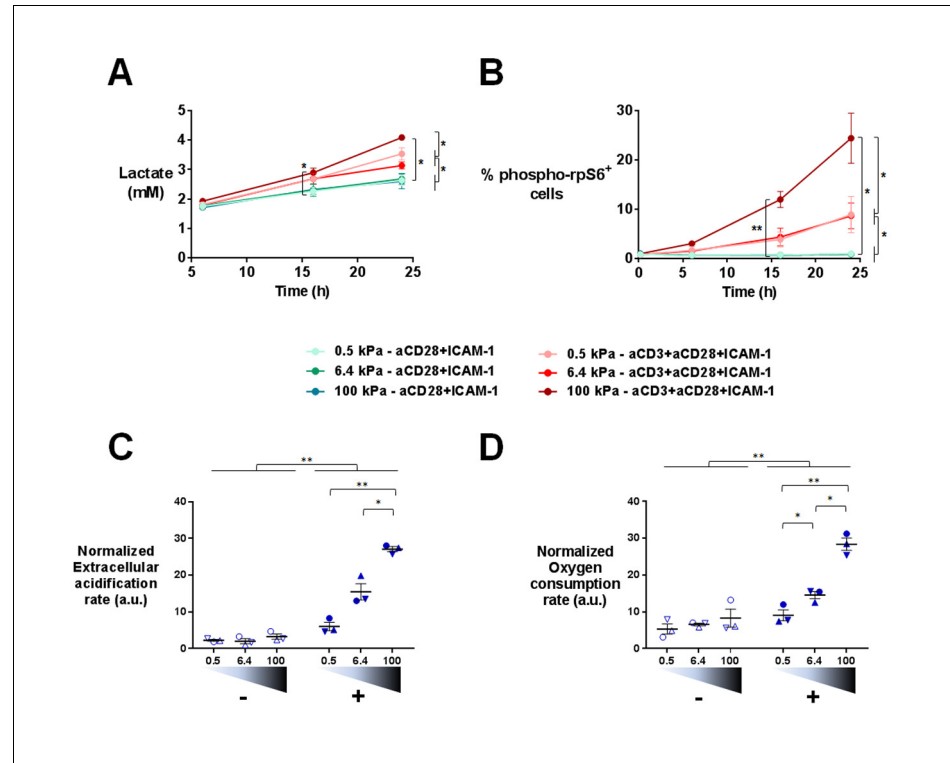

**Figure 5.** T cell metabolism is modulated by increased stiffness. (**A**) Lactate production in the supernatant of T cell cultures on PA-gels of varying stiffness. ($n_{Donors}$: 3). (**B**) Percentage of phospho-rpS6$^+$ T cells cultured on PA-gels of varying stiffness ($n_{Donors}$: 4). (**C**) Overall glycolytic capacity of T cells cultured on PA-gels of varying stiffness for 48 hr. The extracellular acidification rate is normalized to the number of cells per well. Mean values with standard error are shown ($n_{Donors}$: 3). (**D**) Maximal mitochondrial respiration of T cells following culture on PA-gels of varying stiffness for 48 hr. The oxygen consumption rate is normalized to the number of cells per well. Mean values with standard error are shown ($n_{Donors}$: 3). For statistical analysis, paired parametric t-tests were performed: \*\*p-value<0.01, \*p-value<0.05.

The following figure supplement is available for figure 5:

**Figure supplement 1.** Phospho-rpS6 and metabolism: additionnal data.

---

hr after activation and stably increased during time (*Figure 5B*, *Figure 5—figure supplement 1A*). Also, the percentage of phospho-rpS6$^+$ T cells did not show any increase when rigidity went from 0.5 to 6.4 kPa, but was significantly increased for substrate of 100 kPa (*Figure 5B*, *Figure 5—figure supplement 1A*). These results correlated with the lactate production, which was only significantly increased by rigidity of 100 kPa.

Following culture on PA-gels, we further analyzed T cell metabolic changes in terms of overall glycolytic capacity and maximal mitochondrial respiratory capacity. While, at 24 hr no clear differences were observed in the glycolytic capacity and maximal respiration of T cells activated on varying stiffness (*Figure 5—figure supplement 1B and C*), both parameters demonstrated a potentiation with stiffness after 48 hr of culture (*Figure 5C and D*). The apparent discrepancy between increased lactate production on the aCD3 coated 100 kPa gel (*Figure 5A*) and the lack of stiffness potentiation at 24 hr might be due to differences in the performed assays. While lactate production was measured in the actual culture medium at specific time points, T cell glycolytic capacity was measured on cells taken out of their culture conditions.

Changes in T cell metabolism are required to fuel proliferation induced upon T cell activation. Transcriptomic analysis indicated that TCR/CD3-induced mitotic cell cycle was sensitive to PA-gel stiffness, that is expression of cell-cycle-related genes, gene sets and pathways was potentiated in T

cells activated by aCD3 on substrates of increasing stiffness (*Figure 3A and B*, *Supplementary files 2–5*). Moreover, nucleotide synthesis related genes were also increased by stiffness (*Supplementary files 2–5*). We thus measured cell cycle progression and proliferation of human T lymphoblasts activated in the same conditions as in the transcriptomic and metabolic analyses. Results revealed that, at 24 hr, cell cycle progression, that is fewer cells in $G_0/G_1$ phases and significantly more cells in S phase and $G_2/M$ phases, was only evident for T cells activated by aCD3 on the high end of the stiffness range (100 kPa) (*Figure 6A*, *Figure 6—figure supplement 1A*). After 72 hr of culture, cell cycle progression was induced by aCD3 within the whole stiffness range tested (*Figure 6B*). Finally, TCR/CD3-induced proliferation of T cells was measured at 72 hr, demonstrating a graded response to stiffness (*Figure 6C*, *Figure 6—figure supplement 1B*).

Altogether, these results show that, in 24 hr, varying stiffness within the range measured for APCs (hundreds to thousands of Pa) does not modulate T cell metabolism or cell cycle progression induced by low density of activating molecules (our conditions herein). Instead, metabolism and cell cycle progression are increased by high stiffness (100 kPa), observed for tissue, extracellular matrix or tumoral environment (*Paszek et al., 2005*; *Cox and Erler, 2011*). These results also show that T cell response to stiffness builds on with time, resulting eventually in a potentiation of T cell proliferation within the whole stiffness range tested.

## T cell activation is potentiated by APC mechanical properties

The results reported so far, displaying potentiation of TCR-induced T cell activation by substrate stiffness, were obtained with PA-gels coated with activating molecules. Previous reports that investigated the effect of substrate stiffness on T cell activation also used artificial substrates such as PA or PDMS (*Judokusumo et al., 2012*; *Hui et al., 2015*; *Tabdanov et al., 2015*; *O'Connor et al., 2012*). Therefore, in order to assay the role of mechanical properties of substrates in a more physiological model, we switched to an APC system.

To obtain APCs of different mechanical properties, we used confluent cultures of adherent HeLa-CIITA cells expressing MHC class II molecules (*Stumptner-Cuvelette et al., 2001*). Confluency was chosen to avoid a direct contact of the T lymphocytes with the PDMS substrate. HeLa-CIITA cells were cultured to confluence for 48 hr on fibronectin-coated PDMS gels of two stiffness values, 1.5 and 28 kPa. Expression of the MHC class II molecule HLA-DR and the adhesion molecule ICAM-1 by HeLa-CIITA cells was the same on both PDMS substrates (*Figure 7—figure supplement 1B*).

It was previously shown that cells grown on fibronectin-coated substrates of varying stiffness, adapted their spreading area (*Wang et al., 2000*; *Georges and Janmey, 2005*; *Solon et al., 2007*), their cell rigidity (*Solon et al., 2007*; *Tee et al., 2011*) as well as their cell tension (*Engler et al., 2006*; *Basu et al., 2016*) to the stiffness of the underlying substrate. We measured HeLa-CIITA cell area following spreading on the fibronectin coated PDMS gels. HeLa-CIITA cells showed more spreading on 28 kPa gels ($569 \pm 25$ $\mu m^2$, $n_{cells}$: 215) than on 1.5 kPa gels ($453 \pm 14$ $\mu m^2$, $n_{cells}$: 254) (*Figure 7A and B*), showing adaptation to stiffness. We also directly measured the Young's moduli of individual HeLa-CIITA cells plated on the different fibronectin-coated PDMS substrates with a custom-made technique based on the Hertz contact theory and similar in principle to atomic force microscopy (*Figure 7—figure supplement 1A*). Although the differences in HeLa-CIITA cell rigidity on the two different PDMS substrates were not significant, the tendency was for a higher Young's modulus for cells plated on the stiffer substrate: $1.72 \pm 0.2$ kPa ($n_{cells}$: 15) on 28 kPa versus $1.43 \pm 0.15$ kPa ($n_{cells}$: 13) on 1.5 kPa (*Figure 7C*). While these values for HeLa cells are in excellent agreement with previous AFM measurements (*Shimizu et al., 2012*), they reveal that HeLa cells modulated their Young's modulus weakly with substrate rigidity as compared to fibroblasts (*Solon et al., 2007*) and mesenchymal stem cells (MSCs) (*Tee et al., 2011*). This weak increase with substrate rigidity might be due to the different cell type used, but also due to the fact that HeLa cells were confluent. For instance, confluent human umbilical vein endothelial cells were shown to spread less and display lower cell rigidity than individual cells (*Stroka and Aranda-Espinoza, 2011*).

Human $CD4^+$ T lymphoblasts were added on the confluent HeLa-CIITA cultures on 1.5 kPa or 28 kPa PDMS gels along with different concentrations of the TSST-1 superantigen. After 24 hr culture, we measured production of IFNγ and TNFα in the supernatant (*Figure 7D and E*) and surface expression of CD25 (*Figure 7—figure supplement 1C*). Addition of TSST1 induced a dose-dependent increase of cytokine production that was higher when the HeLa-CIITA APCs were plated on the stiffer 28 kPa gel than on the softer 1.5 kPa gel. Expression of CD25 did not show any modification

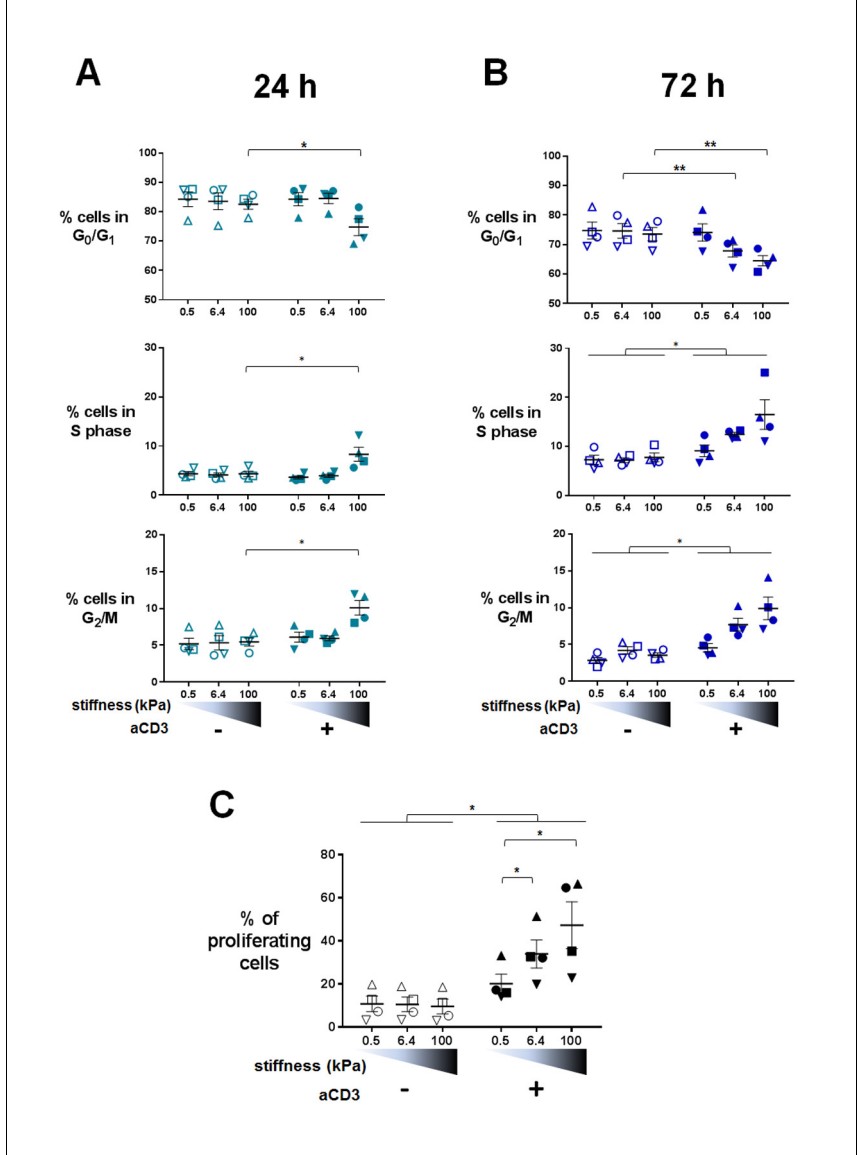

**Figure 6.** Proliferation and cell cycle progression are potentiated by stiffness in response to TCR/CD3 induced activation. The percentages of cells in $G_0/G_1$, S phase and $G_2/M$ are shown for (A) 24 hr ($n_{Donors}$: 4) and (B) 72 hr ($n_{Donors}$: 4). (C) Percentage of proliferating T cells following 72 hr culture on PA-gels of varying stiffness. ($n_{Donors}$: 4). Mean values with standard error are shown. For statistical analysis, paired parametric t-tests were performed: **p-value<0.01, *p-value<0.05.

The following figure supplement is available for figure 6:

**Figure supplement 1.** Cell cycle and proliferation: additionnal data.

in the different stiffness conditions (*Figure 7—figure supplement 1C*). These results are consistent with results obtained when T cells were cultured on PA-gels of different stiffness, that is cytokine secretion is potentiated by the stiffness of aCD3-coated gels (*Figure 4*).

It is worth noting that, in the co-culture setting, T lymphocytes were not submitted to the stiffness range of the PDMS gels (1.5 to 28 kPa) but rather to the small variation of mechanical properties of the HeLa-CIITA monolayers (~ hundreds of Pa), suggesting that cytokine production by T lymphocytes is exquisitely sensitive to stiffness.

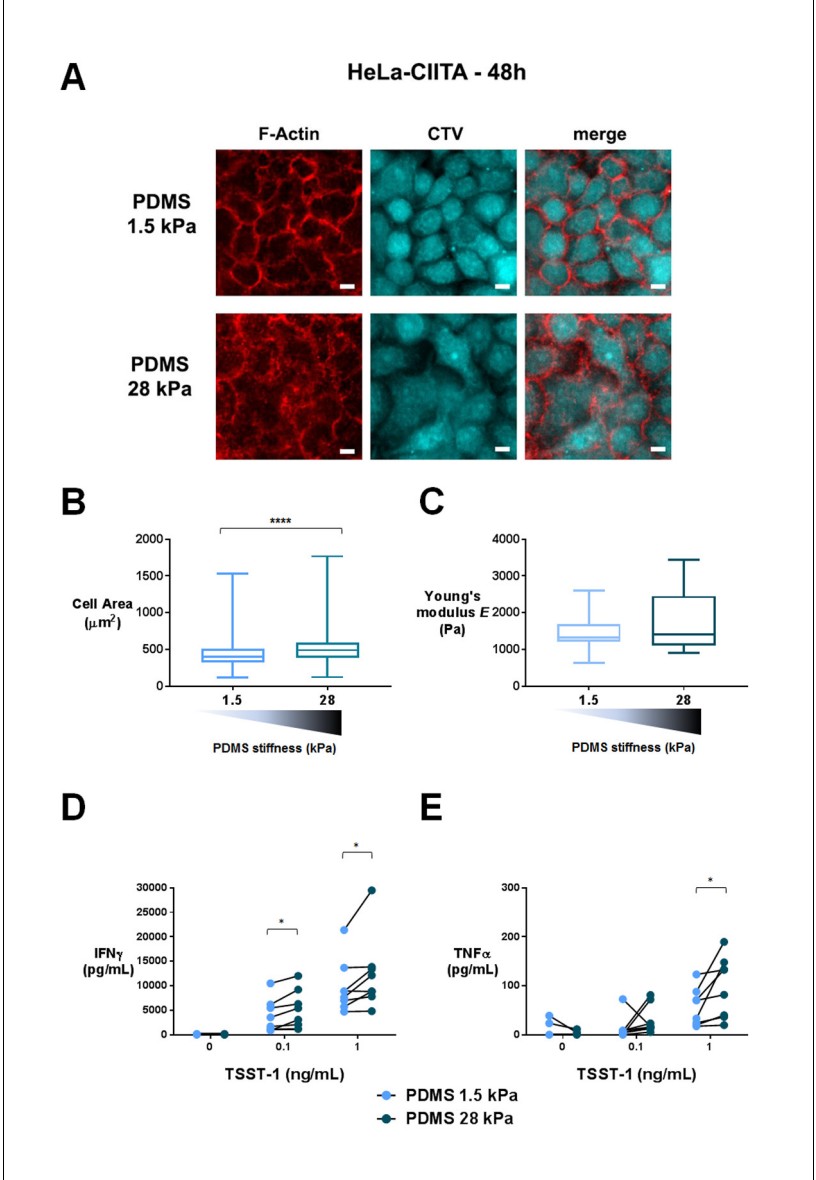

**Figure 7.** T cell activation is potentiated by APC mechanical properties. (**A**) HeLa-CIITA cells were grown at confluence on fibronectin coated PDMS gels of varying stiffness and were stained with phalloidin (F-Actin, in red) and cell trace violet (CTV, in cyan) (scale bar: 10 μm). (**B**) Area of HeLa-CIITA cells cultured on 1.5 kPa (453 ± 14 μm$^2$, $n_{cells}$ = 254) and 28 kPa (569 ± 25 μm$^2$, $n_{cells}$ = 215) PDMS gels. Boxes and whiskers for minimum and maximum are shown. For statistical analysis, unpaired t-tests with Welch's correction were performed: ****p-value<0.0001. (**C**) Young's modulus of HeLa-CIITA cells cultured on 1.5 kPa (1.43 ± 0.15 kPa, $n_{cells}$ = 13) and 28 kPa (1.72 ± 0.2 kPa, $n_{cells}$ = 15) PDMS gels. Boxes and whiskers for minimum and maximum are shown. Production of D) IFNγ and E) TNFα by T cells interacting with HeLa-CIITA on PDMS gels of varying stiffness in the presence of different TSST-1 superantigen concentrations. The response for individual donors is shown ($n_{Donors}$: 8). For statistical analysis, paired parametric t-tests were performed: *p-value<0.05.

The following figure supplement is available for figure 7:

**Figure supplement 1.** Characterization of APC mechanical properties and T cell activation: additionnal data.

We tried to increase the rigidity of HeLa-CIITA APCs by growing them to confluence on glass slides (rigidity in the order of GPa). In this condition, HeLa-CIITA did not show any increase in cell area as compared to cells grown on 28 kPa (*Figure 7—figure supplement 1D*). This might reflect the fact that cell rigidity had already reached saturating conditions, as shown for fibroblast and MSC rigidity on substrate stiffness values of 20 kPa and above (*Solon et al., 2007*; *Tee et al., 2011*). The lack of increase in HeLa-CIITA cell area on glass was accompanied in co-cultures by a lack of increase in TCR-induced cytokine production by T lymphoblasts (*Figure 7—figure supplement 1E*).

These results demonstrate that T cell activation by adherent APCs plated on substrates of varying stiffness follows the same rules as activation on artificial substrates, that is cytokine production is enhanced by increasing stiffness. Moreover, we find that TCR mechanosensitivity also occurs when T cells are activated with the superantigen TSST1, showing that this sensitivity is not induced only by aCD3. Altogether, these results suggest that APC mechanical properties can be an important physiological parameter contributing to antigen-induced cytokine production by CD4$^+$ T cells.

## Discussion

Understanding how CD4$^+$ T cells respond to the mechanics of their environment is critical in order to decipher the modulation of the adaptive immune responses by cells and tissues in normal and pathological conditions. T cell responses can be directly affected by various biophysical cues of the environment (*Hivroz and Saitakis, 2016*; *Husson et al., 2011*; *Judokusumo et al., 2012*; *O'Connor et al., 2012*; *Hui et al., 2015*; *Tabdanov et al., 2015*; *Hu and Butte, 2016*). One of the most important mechanical parameters is substrate stiffness, which can affect cellular responses and eventual cell fate (*Gasiorowski et al., 2013*; *Discher et al., 2005*). In this work, we extended current understanding of mechanical modulation of immune responses by looking into the effect of a physiological range of stiffness values, corresponding to activated dendritic cells, endothelial cells and tissues, on a variety of human T cell functions. Overall, our data reveal that stiffness influences in a rheostat-like fashion many aspects of CD4$^+$ T cell response, that is spreading, stop signal, transcription of several genes, cytokine production, metabolism and cell cycle progression.

Stiffness affects ICAM-1 dependent migration of human CD4+ T lymphoblasts. Indeed, mean instantaneous velocities and travelled distances are increased for T cells on ICAM-1 coated 100 kPa gels as compared to 0.5 and 6.4 kPa gels (*Figure 1A and B*). These results correlate with the fact that mechanically induced transitions of LFA-1/ICAM-1 binding promoted T cell migration (*Morin et al., 2008*; *Jacobelli et al., 2009*) and APC scanning (*Comrie et al., 2015*). Variation of migration parameters by stiffness displayed an initiation threshold between 6.4 kPa and 100 kPa. This might be due to the fact that force transduction depends on the mechanical properties of the actin–talin–integrin–ligand clutch which is triggered by talin unfolding above a stiffness threshold (*Elosegui-Artola et al., 2016*). Below this threshold, integrins unbind before talin can unfold, while above the threshold, talin unfolds and binds to vinculin, leading to adhesion and cell activation. Whether this is the same mechanism that occurs in T cells would require further testing using the talin (*Kumar et al., 2016*) or LFA-1 (*Nordenfelt et al., 2016*) tension sensors recently described. Stiffness also influenced the TCR-induced stop signal of T cells (*Figure 1C*). This is probably related to the mechanotransducing function of TCR/CD3, as will be discussed in the following paragraphs.

These findings have important physiological implications. Indeed, tissue or target cell stiffness can change in pathological conditions. Inflammation induces cell stiffness (*Mathur et al., 2001*; *Kataoka et al., 2002*; *Bufi et al., 2015*). Tumors (*Paszek et al., 2005*) and metastatic lymph nodes (*Choi et al., 2015*) have been shown to be much stiffer than normal tissue. Furthermore, infections and TNF-mediated signaling have been shown to induce arterial stiffness (*Dulai et al., 2012*; *Park and Lakatta, 2012*; *Evani et al., 2016*). In these pathological conditions, tissue stiffness might play a role in modulating the efficiency of T cell scanning. On the other hand, metastatic cancer cells and leukemic cells were reported to be generally very soft (*Rosenbluth et al., 2006*; *Lam et al., 2007*; *Bufi et al., 2015*), in the range of 0.05 to 0.2 kPa, which could be one more mechanism of immune evasion. Therefore, stiffness of the environment is probably a key regulator of T cell migration and antigen detection in vivo.

One of the striking features revealed by our study is the morphology adopted by T cells on substrates of physiological stiffness bearing aCD3 (*Figure 1D*). Scanning electron microscopy images of T cells interacting with substrates from 0.5 to 100 kPa reveal that, in contrast to glass substrates, T

cells do not spread evenly but rather form small protrusions, which tend to spread more on stiffer substrates (*Figure 1D*). This is consistent with in vitro studies showing that T cells probe their environment through close contacts over small areas (around 0.2 µm²) (*Brodovitch et al., 2015*). Moreover, the structures described herein resemble the so-called 'invadosome/podosome-like protrusions' (ILPs), developed by effector/memory T cells when probing endothelium and APCs (*Sage et al., 2012*; *Kumari et al., 2015*). These protrusions might constitute small sensory organelles able to scan the mechanical properties of the substrate (*Albiges-Rizo et al., 2009*; *Martinelli et al., 2014*) and to develop forces, which adapt to stiffness and induce local membrane deformation. This deformation may in turn induce lateral movement of CD45, because of its rigid extracellular domain (*Chang et al., 2016*), leading to its exclusion from the tip of the protrusion and to induction of TCR signaling, which would be consistent with the kinetic segregation model for TCR triggering (*Davis and van der Merwe, 2006*; *van der Merwe and Dushek, 2011*). As shown in our study, stiffer substrates would cause more deformation of cellular components at the surface and better exclusion of CD45 leading to increased numbers of successful TCR/pMHC interactions. Moreover, the mechanical stress imposed by the development of these ILPs on the TCR/pMHC bonds may induce TCR conformational changes leading to signaling (*Lee et al., 2015*). Since ILPs are controlled by the actin cytoskeleton (*Sage et al., 2012*; *Kumari et al., 2015*), further work looking into actin cytoskeleton control and dynamics on substrates of varying stiffness will be required to elucidate this mechanism.

Stiffness had a rheostat-like effect on most of the T cell functions tested. Which could be the mechanisms underlying such potentiation? Our results are reminiscent of studies showing the graded response of TCR signaling to the density of pMHCs and/or their affinity for TCR. Indeed, initial spreading rate of T lymphocytes (*Brodovitch et al., 2015*), stop signal (*Skokos et al., 2007*; *Moreau et al., 2012*), gene expression profiles (*Gottschalk et al., 2012*; *Guy et al., 2013*; *Tkach et al., 2014*; *Allison et al., 2016*), cytokine production, proliferation (*Hemmer et al., 1998*; *Zehn et al., 2009*; *Corse et al., 2010*) and metabolic remodeling (*Rabinowitz et al., 1996*) were shown to be strongly dependent on the concentration and affinity of the encountered TCR ligand. We propose that the stiffness of substrates bearing TCR ligands can act as a modulator of TCR ligand avidity. Indeed, forces exerted on agonist pMHC/TCR bonds were shown to prolong lifetimes of these bonds; this has been called the catch bond effect (*Liu et al., 2014*; *Hong et al., 2015*). Furthermore, stiffness was shown by others and us to modulate the forces developed by T cells (*Husson et al., 2011*; *Bashour et al., 2014*). Thus, increased forces developed by CD4⁺ T cells on the TCR/ligand bonds present on stiff substrates should increase avidity of the bonds, leading to potentiation of TCR/CD3-induced activation of T cells. Therefore, our findings show that the physiological parameters promoting TCR-induced T cell responses include not only the number of ligands and the TCR affinity for ligand, but also the stiffness of the surface presenting the ligands. It is also worth noting that, in this study, low densities of activating molecules, which probably mimic the physiological density of agonist pMHCs on APCs, were used. Results for higher agonist densities might differ as shown for other cell types (*Engler et al., 2004*). Since mechanics was proposed to assist in the discrimination of antigen (*Liu et al., 2014*; *Hong et al., 2015*), it would be particularly interesting to test the interplay of different ranges of pMHC density and/or affinity and substrate stiffness.

Beyond the overall rheostat-like effect of stiffness in TCR-mediated signaling, our results show that different functions have variable sensitivity to stiffness. While most of the genes showed higher increase for the transition from the 0.5 kPa to 6.4 kPa, it is worth noting that many genes showed high sensitivity to mechanical load, displaying a continuous increase in expression induced by few hundreds of Pa to one hundred kPa (*Figure 3D*, points close to the diagonal). This was particularly true for several cytokines (at the transcription and protein level) and was confirmed using co-cultures with model APCs with varying mechanical properties. These results demonstrate that the TCR is a highly sensitive mechanosensor, which can discriminate between small variations of stiffness values reported for APCs (few hundreds of Pa). In contrast, other TCR-induced functions are less sensitive to stiffness of the substrate. T cell arrest was not as sensitive in the same range (0.5 to 6.4 kPa), whereas it was increased for stiffer substrates (100 kPa). Furthermore, within the first 24 hr, metabolic remodeling (*Figure 5A*, *Figure 5—figure supplement 1B and C*), phosphorylation of the rpS6 ribosomal protein (*Figure 5B*, *Figure 5—figure supplement 1A*), and cell cycle progression (*Figure 6A*) were only increased for the stiffest value tested (100 kPa). This suggests that, early on

stiffness modulates TCR-induced activation only in extreme conditions, such as pathological increase of tissue stiffness. Yet, at later time points (48 hr, 72 hr), metabolic remodeling, cell cycle progression and proliferation were modified for the whole range of stiffness tested (500 Pa to 100 kPa) (*Figure 5C,D, and Figure 6B and C*) suggesting that response to stiffness builds on with time. Such latency may be related to amplification loops induced by cytokines, which require time to be produced. Indeed, signaling pathways induced by cytokine can add up to TCR signaling for an integrated T cell metabolic response and cell division (*Marchingo et al., 2014*). Along this line, it is worth noting that the Jak/Stat signaling pathway induced by cytokines was gradually increased by stiffness (*Supplementary files 2–4*).

The mechanisms involved in TCR mechanosensing are yet to be unraveled. In this study, we show that stiffness also regulates the expression of lamins by activated T cells (*Figure 2C*). This family of molecules has been shown to control cell trafficking and differentiation of hematopoietic cell types (*Shin et al., 2013*), as well as T cell activation (*González-Granado et al., 2014*). Expression of lamins has also been shown to scale with tissue rigidity and to control stem cell differentiation directed by matrix stiffness (*Swift et al., 2013*). It is thus possible that lamins would transduce the mechanical signal from substrate stiffness to the nucleus and thus regulate the observed differences in T cell gene expression. This would require further testing using conditional knockouts or silencing of lamin genes.

Overall, our results suggest that cell and tissue stiffness is a general key regulator of T cell responses controlling effector functions of CD4$^+$ T cells. This mechanical tuning of T cell activation could be particularly important in vivo for locating and responding to scarce agonist pMHCs. Finally, substrate stiffness should be considered for the optimal expansion of T cells used in applications such as cancer immunotherapy.

# Materials and methods

## Cell culture

Mononuclear cells were isolated from peripheral blood of healthy donors on a ficoll density gradient. Human total and memory CD4$^+$ isolation kits (Miltenyi Biotech, Bergish Gladbach, Germany, cat. no. 130-096-533 and 130-091-893 respectively) were used for the purification of T cells. To obtain lymphoblastoid effector T cells (*Larghi et al., 2013*), six-well plastic plates were coated overnight at 4°C with aCD3 (OKT3 clone, eBioscience, cat. no. 16-0037-85, 2.5 µg/mL final concentration in 1.3 mL). Wells were washed and 5.4 × 10$^6$ purified primary human total CD4$^+$ T cells were then plated per well in the presence of soluble anti-CD28 (LEAF Purified anti-human CD28 from CD28.2 clone, Ozyme, cat. no. BLE302923) at 2.5 µg/mL final concentration and recombinant human IL-2 (20 U/mL) (Novartis, Basel, Switzerland) in RPMI culture medium (Life Technologies, Carlsbad, CA) with 10% fetal calf serum (FCS). Fresh medium containing human recombinant IL-2 (20 U/mL) was added every 3 days and lymphoblasts were used from day 6. Following 24 hr cultures on PA-gels or PDMS gels or glass coverslips, T cells were recovered for flow cytometry, gene expression and metabolic analysis and the supernatant was tested by ELISA kits for the production of cytokines IFNγ, TNFα and IL-2 (OptEIA, BD Biosciences, San Jose, CA, cat. no. 550612, 550610 and 550611 respectively).

HeLa cells transfected to express the CIITA transcription factor (*Stumptner-Cuvelette et al., 2001*) were cultured in DMEM medium (Life Technologies, Carlsbad, CA) supplemented with 10% FCS. For selecting CIITA expressing cells, cultures were also supplemented with hygromycin B (Life Technologies, Carlsbad, CA, cat. no. 10687010) at a final concentration of 0.3 mg/mL. To harvest cells, a 10 min incubation with trypsin containing medium (TrypLE, Life Technologies, Carlsbad, CA, cat. no. 12605010) was used. Cells were free of mycoplasma as revealed by the MycoAlertTM Mycoplasma detection kit (Lonza, Basel, Switzerland, cat. no. LT07-710).

## Hydrogel preparation for cell culture

Poly-acrylamide hydrogels (PA-gels) were prepared with different acrylamide (Sigma-Aldrich, Saint Louis MO, cat. no, A9099) and bis-acrylamide (Sigma-Aldrich, cat. no. M7279) composition (*Table 1*). Round glass coverslips (diameter 12 mm, 14 mm, 15 mm) (VWR, Radnor, PA). were prepared as follows: a) sonication for 5 min in isopropanol, b) washing with double distilled H$_2$O and ethanol and drying, c) O$_2$ plasma etching (2 min) with a Pico Low-pressure Plasma System (Diener, Ebhausen,

Germany) to render them hydrophilic. The glass coverslips were then functionalized with a 1% Bind-Silane (Sigma-Aldrich, cat. no. M6514) solution in ethanol-10% acetic acid for 20–30 min at room temperature. The coverslips were then washed with ethanol and dried. Solutions containing acrylamide, bis-acrylamide and streptavidin-acrylamide conjugate (Life Technologies, cat. no. S21379) (used at a 1/100000 molecular ratio to acrylamide) were prepared at the desired volume and concentration: (*Table 1*). Prior to polymerization, the solutions were degassed under vacuum for 5 min. Following the activation with TEMED (Sigma-Aldrich, cat. no. T9281) and ammonium persulfate (Sigma-Aldrich, cat. no. A3678), 22 µL of the polymerization mix were rapidly added on a non-functionalized 18 mm diameter coverslip and a Bind-Silane functionalized glass coverslip (12, 14 or 15 mm diameter) was added on top. Polymerization was performed under constant supply of argon gas for 2 hr. The surface tension of the liquid mix ensured the formation of a uniform gel layer. The PA-gel-coated coverslips were then separated from the non-functionalized coverslips, washed and stored in PBS until further use. The 12 mm diameter coverslips were used for scanning electron microscopy, the 14 mm ones were used for video microscopy and the 15 mm were used for cell culture with regular plastic 24-well plates. Non-gel-coated glass coverslips were coated overnight with neutravidin (Life Technologies, cat. no. 11544037) (200 µg/mL in PBS). Commercial PDMS gels were also used at two specific stiffness values, 1.5 and 28 kPa (Ibidi, Martinsried, Germany, cat. no. 81291 and 81191 respectively).

## Mechanical properties of PA-gels

Mechanical properties of the gels were determined using a SR500 shear rheometer (Rheometrics, Piscataway, NJ) with a plate-plate cell of 25 mm in diameter. The shear storage modulus G´ was measured at a strain of 5%, for two frequencies (1 rad/s and 5 rad/s). G´ of PA-gel samples of typically 1 mm thickness and 40 mm in diameter were measured when decreasing the gap, that is distance between the plates of the rheometer (*Figure 1—figure supplement 1A*). G´ increased as the gap was decreased indicating progressive loading of the sample. We retained the maximum value of G´ as function of the gap as the characteristic shear modulus of the sample. The tensile elastic modulus $E$ (Young's modulus) was retrieved using: $E = 2 (1+\nu) \times G´$, where we took $\nu = 0.45$ for the Poisson's ratio of poly-acrylamide (*Takigawa et al., 1996*). The Young's modulus values, which are a measure of the stiffness of PA-gels, are shown in *Table 1*. At least three different gels from three different preparations were measured. The presence of streptavidin-acrylamide did not alter the mechanical properties of PA-gels.

## Functionalization of surfaces

We employed specific biotin-streptavidin binding for the functionalization of PA-gels and neutravidin-coated glass coverslips. A total biotinylated protein amount of 10 µg/mL was used for each coating strategy. All surfaces were incubated overnight with the various biotinylated proteins in PBS-BSA-2% at 4°C. The following molecules were used: biotinylated mouse anti-human CD3ε (OKT3 clone, EXBIO, Praha, Czech Republic); biotinylated mouse anti-human CD28 (CD28.2 clone, EXBIO); human ICAM-1/Fc chimeric protein (R and D Systems, Minneapolis, MN, cat. no. 720-IC-200) biotinylated with the Sulfo-NHS-EDC biotin kit (Thermo Scientific, Waltham, MA, cat. no. 21327). The coating with biotinylated proteins was checked using the following fluorescently labeled antibodies: a) for biotinylated mouse-derived antibodies: anti-mouse IgG Fab$_2$ fragment conjugated with PE (Jackson Immunoresearch, West Grove, PA, cat. no. 115-116-146), and b) for biotinylated ICAM-1: anti-human IgG Fab$_2$ fragment conjugated with PE (Jackson Immunoresearch, cat. no. 109-116-098). We selected the PE-conjugated antibodies in order to specifically detect the biotinylated proteins on the surface of PA-gels, since the bulky phycoerythrin would not be able to enter the nanometer-sized pores (*Trappmann et al., 2012*) of even the softest gel used. Immunofluorescence experiments revealed that the antibodies are concentrated on the top surface of PA-gels, as viewed from z-axis projection. In order to adjust for similar antibody coating, the amount of streptavidin-acrylamide molecules in the 6.4 and 100 kPa gels is higher, as mentioned above and in the literature (*Judokusumo et al., 2012*). In that case, biotinylated protein coating is similar for PA-gels of all three different stiffness values (*Figure 1—figure supplement 1C*). For neutravidin-coated glass coverslips, we also used a non-specific biotinylated rat isotype IgG (BD Biosciences, cat. no. 553983) at a 10x molar excess in order to occupy excess biotin-binding sites, yielding a coating efficiency similar

to PA-gels ('Glass Neutr. 1/10', *Figure 1—figure supplement 1C*). PDMS gels were coated overnight with 2.5 µg/mL of fibronectin (Sigma Aldrich, cat. no. F2006) in PBS. For cell culture: $4 \times 10^5$ T cells were added on PA-gel or glass coverslips for 24 hr; and HeLa-CIITA cells were cultured to confluence for 48 hr on the fibronectin-coated PDMS gels before adding T cells ($5 \times 10^5$ per plate) for additional 24 hr. Untreated or biotinylated (with the above kit) fibronectin was also used to test non-specific versus specific binding on PA-gels with the following antibodies: anti-human fibronectin rabbit IgG (Sigma Aldrich, cat. no. F3648) and anti-rabbit IgG Alexa-Fluor 546 (Life Technologies, cat. no. A-11035)(*Figure 1—figure supplement 1B*).

A theoretical surface density of immobilized molecules was calculated. To do so, we made three assumptions: 1) The volume of the hydrated gel (swollen gel in culture medium) that we can calculate from its thickness and the diameter of the slide used, is 40% bigger than the initial volume of the polymerization mix (this was based on a previous study [*Hynd et al., 2007*]); 2) All the streptavidin-acrylamide molecules in the polymerization mix did polymerize in the gel; 3) Biotinylated Abs and ICAM-1/Fc molecules can only access the first 10 nm of the gels (10 nm being the approximate size of the streptavidin molecule) due to the size of the pore reported in the literature (15 nm for PA-gels of 0.5 kPa and smaller for the more rigid gels [*Trappmann et al., 2012*]). According to these assumptions, we calculated the density of immobilized molecules. For the PA-gels used in cell cultures, 22 µL of the polymerization mix was added in between two glass coverslips, one silane-functionalized (15 mm diameter) and one non-functionalized (18 mm diameter). The gel formed between the glass coverslips but also formed on the extended surface of the larger 18 mm diameter coverslip. From the measured (by microscopy) gel thickness of the re-hydrated PA-gels (25–50 µm) and the diameter of the slide (15 mm), we calculated the volume of the hydrated gels: 4.4–8.8 µL. We assumed that the hydrated PA-gels have a ~ 40% bigger volume than their initial polymerization mix (*Hynd et al., 2007*), thus the initial volume of the polymerization mix between the coverslips was ranging from 3 to 6 µL. Assuming that all the streptavidin-acrylamide in the initial polymerization mix polymerized, we calculated the amount of molecules of streptavidin-acrylamide in this initial volume. To determine the surface density of streptavidin-acrylamide, we assumed that biotinylated Abs and ICAM-1/Fc molecules could only access the streptavidin in the first 10 nm of the gels, which corresponds to the approximate diameter of a streptavidin molecule (*Neish et al., 2002*) and also, all these molecules would be available for binding with soluble biotinylated antibodies taking into account the gel pore size (~15 nm). We thus calculated the effective surface density of streptavidin-acrylamide within a range of 20 to 40 molecules/µm². Since in most of the experiments a molar ratio of 10 anti-CD28+ICAM-1/Fc to 1 anti-CD3 was used, the theoretical density of anti-CD3 is of 2 to 4 molecules/µm² for 20 to 40 anti-CD28+ICAM1-Fc molecules/µm² for the 0.5 kPa gels. Finally, since the overall protein coating of PA-gels, measured by microscopy, was similar on the gels of different rigidities (*Figure 1—figure supplement 1C*), this range of values for surface density applies to all PA-gels used in the study.

## Video microscopy

T cell migration on gels was monitored either with a Nikon TiE video microscope (Nikon, Tokyo, Japan) equipped with a cooled CCD camera (HQ2, Photometrics, Tucson, AZ) or a Nikon TE2000 microscope with a Cascade 1K camera (Photometrics). Experiments were performed in culture medium and transmission movies were acquired using a 10x or 20x objective, with one frame every 5 s. Analysis was performed using Fiji (*Schindelin et al., 2012*) and R software. First, T cells were tracked for 5 min using the TrackMate plugin of Fiji. Then, track representation and analysis were obtained with R. For each T cell's track, we computed the mean instantaneous velocity and the maximum distance traveled on the substrate, which is the maximum distance separating two points within a single track. We considered a T cell as arrested when its maximum distance traveled on the substrate was less than 10 µm for the 5-min tracks (<2 µm/min) (*Jacobelli et al., 2009*). Statistical analysis was performed with GraphPad Prizm six or Origin eight software.

## Flow cytometry for T cell activation and cell cycle

Following cell culture, the following molecules were used to stain cells: Live/Dead fluorescent dye (Life Technologies, L23101) to detect cell viability; CellTrace CFSE (Life Technologies, C34554) or Violet (Life Technologies, C34557) to detect cell proliferation; anti-CD25-PE and anti-CD69-PE (BD

Pharmingen, San Diego, CA, cat. no. 555432 and 555531 respectively) to detect T cell activation; anti-HLA-DR-PE and anti-CD54-APC (BD Pharmingen, 347401 and 559771 respectively) to measure APC surface markers on HeLa-CIITA cells. To identify cell cycle stages, we fixed T cells in 70% ethanol on ice and used a combined treatment of RNAse incubation and propidium iodide staining (Sigma Aldrich, cat. no. R6148 and P4864 respectively) as previously reported (*Vivar et al., 2009*). To measure the amount of phospho-rpS6 ribosomal protein, cells were fixed in Cytofix/Cytoperm solution (BD Biosciences, 554722), permeabilized with Perm/Wash buffer (BD Biosciences, 554723) and stained with an anti-phospho-rpS6(Ser235/236) antibody (Cell Signaling Technology, Danvers, MA, cat. no. 2211L). All experiments were performed with the MACSQuant flow cytometer (Miltenyi Biotech, Bergish Gladbach, Germany).

## Scanning electron microscopy

Scanning electron microscopy was performed for T cells on PA-gels of varying stiffness coated with aCD3+aCD28+ICAM-1 following incubation for 30 min at 37°C. The samples were washed in phosphate buffer (PB, 200 mM, pH 7.4), fixed overnight at 4°C in PB + 2% glutaraldehyde, and finally washed in PB. Samples were then dehydrated by passing through a graded series of ethanol solutions, then dried by the $CO_2$ critical-point method (Quorum Technologies, Guelph, ON, cat. no. CPD75) and coated by sputtering with a 20–40 nm gold thin layer using a Scancoat Six (HHV, Bagnalore, India) sputter coater. Acquisitions were performed using a Gemini SEM 500 (Zeiss, Jena, Germany) at a tilt of 45 degrees.

## mRNA microarrays and data analysis

Following 24 hr cultures on PA-gels of varying stiffness, polysomal RNA was isolated from $6 \times 10^5$ T cells from each condition using the NucleoSpin RNA kit (Macherey-Nagel, Dueren, Germany, cat. no. 740955.50). For each condition, 100 ng of RNA was transcribed into cDNA. Labeled DNA was then hybridized on the Affymetrix human Gene ST2.1, an oligonucleotide microarray, and processed on an Affymetrix GeneTitan device (Affymetrix, Santa Clara, CA). This analysis was performed for four different human donors. Microarray data were then processed into the open source software R (version 3.1.0, www.r-project.org) using packages from Bioconductor (www.bioconductor.org). CEL files with raw data were used and the quality control analysis was performed using ArrayQualityMetrics package (*Kauffmann et al., 2009*). The raw data were preprocessed using the RMA method available in the oligo package (*Carvalho and Irizarry, 2010*). Probes with no annotation were removed from analysis. Finally, the resulting matrix comprised data for 53617 probes.

The PCA of T cell profiles was performed using the Ade4 package (*Dray and Dufour, 2007*) of the R software on all probes, except the not annotated ones. The barycenters were computed from the set of observations in each condition and projected into the PCA plot. Confidence ellipses (95% of confidence) around the barycenters of conditions were then drawn.

Identification of differentially expressed genes was performed with the limma package (*Smyth, 2005*) by computing moderated t-tests. We considered a gene as differentially expressed if its adjusted p-value using the Benjamini-Hochberg method was lower than 0.05.

The gene expression data matrix was filtered to remove genes with low or no expression. We considered a gene as expressed when the mean value of its log expression in all conditions was above 3. The filtered matrix had 11416 probes. Hierarchical clustering by K-means clustering of gene expression profiles was then performed using the ExpressCluster 1.3 tool (http://cbdm.hms.harvard.edu/LabMembersPges/SD.html) with default parameters. The number of iterations was fixed at 1000 and the number of of generated clusters was fixed to 3, in order to provide clear separation of gene expression profiles. Comparison of relative gene expression for the strongly up-regulated cluster was performed with Multiplot (version 1.5.20) (Tempero Pharmaceuticals, Cambridge, MA).

## Gene set enrichment analysis

Data from the mRNA microarrays were used to perform pair-wise comparisons (presence vs. absence of aCD3 for each stiffness value) with the gene set enrichment analysis (GSEA) method (www.broad.mit.edu/gsea). Statistical analysis was performed evaluating nominal p-values and false discovery rates (FDR) based on 1000 permutations. We used the Gene Ontology – Biological Processes (GO-BP) and the Kyoto Encyclopedia of Genes and Genomes (KEGG) pathway gene sets as ranked data

sets. Results were considered significant when the p-value was lower than 0.05 and the FDR was lower than 0.25, according to developer's instructions (*Subramanian et al., 2007*). The GSEA output is mainly characterized by two parameters: the normalized enrichment score (NES) and the false discovery rate (FDR). NES represents the number and differential expression intensity of genes enriched in the corresponding gene set. We used a cut-off NES value of 1.75 for highly significant enrichment (*Supplementary file 2*).

## Pathway analysis

Pathway analysis of differentially expressed genes and gene clusters was performed with the publicly available tool EnrichR (http://amp.pharm.mssm.edu/Enrichr) that provides access to various gene-set libraries, including the GO-BP and the KEGG databases, and computes enrichment of specific pathways (*Chen et al., 2013*; *Kuleshov et al., 2016*). We considered pathways as enriched if their adjusted p-value was lower than 0.05 and ranked them with it.

## Metabolic assays

To monitor lactate production by T cell cultures on PA-gels, the culture supernatant was taken at different time points (6, 16 and 24 hr) and was deproteinized with 10 kDa MWCO spin filters (GE Healthcare, Chicago, IL, cat. no. 28-9331-02 AB) to remove lactate dehydrogenase. Lactate was then measured with a colorimetric assay (Sigma Aldrich, MAK064).

Following a 48 h cell culture on PA-gels, $1.5 \times 10^5$ T cells from each condition were plated on a 96-well XFe extracellular flux analyzer (Seahorse Bioscience, Santa Clara, CA). Changes in extracellular acidification rate and in oxygen consumption rate were then measured according to manufacturer's instructions. From the real-time curves, we calculated the overall glycolytic capacity as well as the maximal mitochondrial respiration. In order to account for growing cell numbers due to proliferation, the measured values were normalized to the final cell count of the 48 hr cultures.

## Measurement of HeLa cell area and Young's modulus

HeLa-CIITA cells cultured for 48 hr on PDMS substrates (1.5 kPa and 28 kPa) and glass were washed with PBS and stained with CellTrace Violet (Life Technologies, C34557). Following washing, the cells were fixed with 4% paraformaldehyde, permeabilized with saponin (0.05% in PBS, Sigma Aldrich, 47036), and stained with phalloidin-Alexa Fluor 647 (Life Technologies, A22287). The samples were then observed on an inverted spinning-disk confocal Nikon TiE microscope (Nikon) equipped with a piezo-stage NanoScanZ (Prior Scientific, Rockland, MA) mounted on a XYZ motorized scanning stage (Marzhauser, Wetzlar, Germany). Three-dimensional stacks of images were acquired with a step of 0.5 mm using a 40x objective and an EM-CCD iXon 897 Andor camera (Andor, Belfast, UK). Metamorph software (Molecular Devices, Sunnyvale, CA) was used for all acquisitions. The cell area was measured from the Z-plane closest to the substrate using ImageJ software.

In order to measure the Young's modulus of HeLa cells cultured on PDMS gels of varying stiffness, we used a custom-made glass cantilever with a spherical tip. HeLa cells in confluent layers were indented with the cantilever, which was calibrated in stiffness, allowing us to measure the indentation force from its deflection. Using the Hertz contact model for a spherical probe (*Sneddon, 1965*), the force-indentation relationship could be written as:

$$F = \frac{4}{3} \sqrt{R} \; E^* \; \delta^{3/2}$$

where $F$ is the indentation force, $R$ the radius of the probe, $\delta$ the indentation depth, and $E^*$ the reduced Young's modulus:

$$\frac{1}{E^*} = \frac{1 - \nu_{cell}^2}{E_{cell}} + \frac{1 - \nu_{probe}^2}{E_{probe}}$$

where $\nu$ is the Poisson's ratio. Since the Young's modulus of the cells is ~kPa while that of the glass probe is ~GPa, $E_{cell} \ll E_{probe}$ and the reduced Young's modulus can be expressed as:

$$E^* = \frac{E_{cell}}{1 - \nu_{cell}^2}$$

For the purpose of comparison with previous measurements on individual HeLa cells with a similar protocol (*Shimizu et al., 2012*), we took $\nu_{cell} = 0.5$. Thus, the force-indentation relationship writes:

$$F = \frac{16}{9} \sqrt{R} \, E_{cell} \, \delta^{3/2}$$

and $E_{cell}$ was retrieved from fitting of the force-indentation curves with R = 5.7 µm. During measurements, Hela monolayers were indented slowly (<0.5 µm/s) a few microns in depth.

The glass probe used here was pulled from 10 cm long – 1 mm diameter glass rods (World Precision Instruments, Sarasota, FL, cat. no. GR100-4), using a PB-7 puller (Narishige, Japan) and its tip was shaped using a MF900 microforge (Narishige, Japan). Its stiffness (3 nN/µm) was calibrated thanks to a reference microplate (*Desprat et al., 2006*).

## Acknowledgements

The authors thank Virginie Bazin (IBPS, Université P. et M. Curie, Paris) for her help obtaining SEM images, and Laurence Bataille, Luigia Pace and Andrés Zucchetti for critically reading the manuscript. This work was supported by funds from ANR (ANR-12-BSV5-0007-01, ImmunoMeca; ANR-13-BSV2-0018 'NeuroImmunoSynapse'), DC-Biol Labex (ANR-10-IDEX-0001–02 PSL* and ANR-11-LABX-0043), 'Who Am I?' Labex (ANR-11-IDEX-0005–02 and ANR-11-LABX-0071) and Fondation pour la Recherche Médicale (FRM, FRM DEQ20140329513).

## Additional information

### Funding

| Funder | Grant reference number | Author |
|---|---|---|
| Agence Nationale de la Recherche | ANR-12-BSV5-0007-01 | Michael Saitakis<br>Stéphanie Dogniaux<br>Nathalie Bufi<br>Atef Asnacios<br>Claire Hivroz |
| Fondation pour la Recherche Médicale | FRM DEQ20140329513 | Michael Saitakis<br>Stéphanie Dogniaux<br>Nathalie Bufi<br>Claire Hivroz |
| Agence Nationale de la Recherche | ANR-10-IDEX-0001-02PSL | Michael Saitakis<br>Stéphanie Dogniaux<br>Christel Goudot<br>Mathieu Maurin<br>Claire Hivroz |
| Agence Nationale de la Recherche | ANR-11-LABX-0043 | Michael Saitakis<br>Stéphanie Dogniaux<br>Christel Goudot<br>Mathieu Maurin<br>Claire Hivroz |
| Agence Nationale de la Recherche | ANR-11-LABX-0071 | Atef Asnacios |
| Agence Nationale de la Recherche | ANR-11-IDEX-0005-02 | Atef Asnacios |
| Agence Nationale de la Recherche | ANR-13-BSV2-0018 | Claire Hivroz |

The funders had no role in study design, data collection and interpretation, or the decision to submit the work for publication.

### Author contributions

MS, Conceptualization, Data curation, Formal analysis, Validation, Visualization, Methodology, Writing—original draft, Writing—review and editing; SD, SA, Validation, Methodology; CG, Software, Visualization, Methodology, Writing—original draft; NB, Formal analysis, Validation, Methodology;

MM, Software, Methodology, Writing—original draft, Writing—review and editing; CR, Methodology; AA, Conceptualization, Resources, Data curation, Formal analysis, Supervision, Funding acquisition, Methodology, Writing—original draft, Writing—review and editing; CH, Conceptualization, Resources, Data curation, Formal analysis, Supervision, Funding acquisition, Writing—original draft, Writing—review and editing

### Author ORCIDs

Claire Hivroz, http://orcid.org/0000-0002-6794-2890

### Ethics

Human subjects: This study was conducted according to the Helsinki Declaration, with informed consent obtained from the blood donors, as requested by the Etablissement Francais du Sang.

## Additional files

### Supplementary files

• Supplementary files 1. Differential Analysis_aCD3 Presence vs absence. Lists of genes with differential expression comparing absence and presence of aCD3 stimulation on PA-gels of varying stiffness and Venn analysis of the lists.

• Supplementary files 2. Gene Set Enrichment Analysis_aCD3 Presence vs Absence. Gene set enrichment analysis results with GO-BP and KEGG databases comparing absence and presence of aCD3 stimulation on PA-gels of varying stiffness and Venn analysis of the results.

• Supplementary files 3. Differential Analysis_GO-BP. Pathway analysis results for differentially expressed genes using GO-BP database and Venn analysis of the results.

• Supplementary files 4. Differential Analysis_KEGG. Pathway analysis results for differentially expressed genes using KEGG database and Venn analysis of the results.

• Supplementary files 5. Up-regulated Cluster. List of genes in the strongly up-regulated cluster and pathway analysis results with GO-BP and KEGG databases.

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
