## [Decision Letter]

Thank you for submitting your article "Mechanical tuning of CD4^+^ T cell responses" for consideration by *eLife*. Your article has been favorably evaluated by Arup Chakraborty (Senior Editor) and three reviewers, one of whom, Michael L Dustin (Reviewer #1), is a member of our Board of Reviewing Editors.

The reviewers have discussed the reviews with one another and the Reviewing Editor has drafted this decision to help you prepare a revised submission.

Summary:

There is significant interest in the role of mechanical forces in T cell activation. The authors use hydrogel substrates with anti-CD3, anti-CD28 and ICAM-1 to stimulate in vitro expanded (>d6) human CD4 T cells. The inclusion of the integrin ligand is a strength, as integrins are mechanotransduction receptors and this distinguishes this study from earlier studies on T cell responses to PA hydrogels. The authors perform cytokine production (0-24 h), motility (8.3 m), cell shape (30 m), gene expression (24 h) and metabolism (24 hr- then measured off substrates) studies on the cells during or after culture on the hydrogels or model antigen presenting cells cultured on different PDMS compositions. While cytokine, motility and gene expression studies generally show a progressive increase in activation parameters with increasing substrate stiffness, the metabolic response of the cells is more consistent across substrate rigidities. The group is very strong in mechanics and TCR signaling, but there are a number of weaknesses in the details of the study which need to be addressed.

Essential revisions:

1) The ligands are all coupled through a biotin-avidin based system to hydrogels. In the Methods the authors discuss using a biotinylated IgG with an irrelevant binding specificity as a "stuffer" to adjust the density of the biologically active components. This is a great idea, but it’s not clear they fully take advantage of it. The authors settle for a 2-fold difference in coating density of ligands between glass and the PA hydrogels. They also don't appear to use the stuffer reagent to keep all ligand densities consistent when, for example, anti-CD3 is omitted. If they did this they should make this clear. Also, in order to compare between studies it would be helpful to calculate the density in molecules/µm2.

2) The speeds for T cell motility on the hydrogels are very fast and the reduction in speed on addition of anti-CD3 is modest – no prior studies would define this as a stop signal. It’s not clear why these values are so high as the frame rate is pretty typical at one frame every 5 seconds. The authors may want to check that the distances or times are calibrated correctly. Even in vivo arrest or stopping would have a threshold of <2 µm/min? Could there be tracking errors that are accounting for centroid displacement every 5 seconds that might lead to an artificial impression that the cells are moving rapidly? How is the cell centroid tracked? The morphologies shown in Figure 4 don't appear consistent with the fast motility values reported for the cells on the stimulatory hydrogels. The cells don't appear sufficiently polarized to move this fast.

3) Data showing T cell cytokine production on APC monolayer (subsection “T cell activation is potentiated by APC mechanical properties”) are striking. However, the stiffness of PDMS used in this section was not comparable to the Young's modulus of PA gels used throughout the studies. PDMS substrates with higher stiffness for cell study can certainly be achieved (See example: Palchesko et al., PLOS One, 2012, doi: 10.1371/journal.pone.0051499). Although the cytokine levels are different in the sample tested. The authors interpreted CD25 expression on APC is insensitive to stiffness in their tested conditions, but whether this expression is "graded" or "all-or-none" in response to the stiffness remains untested. It would be helpful if the authors include PDMS gels with higher stiffness such that at least three data points could be tested to provide a relationship. This would also be of interest in relation to O'Connor et al., 2013 (cited) where antibodies coated on stiffer PDMS hydrogels were less stimulatory than antibodies on a softer (50-100 kPa) substrate. It would be interesting if this optimum at intermediate PDMS stiffness is also detected here when an APC is inserted between the T cell and PDMS substrate.

Ideally, measurement of the local stiffness of the HeLa cells on the PDMS should be provided. The assumption is that the greater spreading of the cells on the stiffer gel means the cells are stiffer, but as other factors could impact the stiffness of the surface this needs to be measured.

4) This study would benefit from better consideration of these time-dependent issues by conducting several of the included experiments at very early (minutes), intermediate (several hours) and late (~24 hours). It is well established that Ag-depended responses emerge over time with differential phases. For example stop signals develop acutely and are reversed over the time frame of tens of minutes to hours. Additionally, while metabolic changes can emerge over hours in a transcriptionally regulated fashion, they can also be altered by diverse (including biomechanical) signals in seconds.

The decision to do the metabolic measurements off the PA substrates weakens them. It’s well known that acid production by T cells is a real time response to antigen recognition (see work from McConnell in late 1990s). So the responses might have been different if they were measured on the respective substrates rather than measured in a plastic well seemingly with no TCR ligands. An earlier time point should probably be selected also.

---

## [Author Response]

*Essential revisions:*

*1) The ligands are all coupled through a biotin-avidin based system to hydrogels. In the Methods the authors discuss using a biotinylated IgG with an irrelevant binding specificity as a "stuffer" to adjust the density of the biologically active components. This is a great idea, but it’s not clear they fully take advantage of it. The authors settle for a 2-fold difference in coating density of ligands between glass and the PA hydrogels. They also don't appear to use the stuffer reagent to keep all ligand densities consistent when, for example, anti-CD3 is omitted. If they did this they should make this clear. Also, in order to compare between studies it would be helpful to calculate the density in molecules/µm2.*

We thank the reviewers for these comments that will help clarify different points of the manuscript.

Concerning the stuffer reagent, we only used it on glass slides (See, [Supplementary-material SD2-data]) because the binding on the glass of neutravidin (thus the number of sites where biotinylated molecules can bind) was so efficient that its density was always higher than the density of streptavidin on the PA gels ([Supplementary-material SD2-data]). For PA gels, as specified in [Supplementary-material SD2-data], we used 10 (for all results except Figure 4—figure supplement 1) to 100 times (Figure 4—figure supplement 1) more anti-CD28 than anti-CD3 and thus considered that omission of anti-CD3 does not drastically change the concentration of anti-CD28+ICAM-1 density on the PA gels.

Regarding the surface density of activating molecules, we were not able to measure it experimentally but we rather calculated a theoretical range of density of streptavidin molecules on the gel surface. To do so we made three assumptions: 1) The volume of the hydrated gel (swollen gel in culture medium) that we can calculate from its thickness and the diameter of the slide used, is 40% bigger than the initial volume of the polymerization mix (based on a previous study (Hynd et al., 2007)); 2) All of the streptavidin-acrylamide molecules in the polymerization mix did polymerize in the gel; 3) Biotinylated Abs and ICAM-1-Fc molecules can only access the first 10 nm of the gels (10 nm being the approximate size of the streptavidin molecule) due to their own size and the size of the pore reported in the literature (15 nm for PA-gels of 0.5 kPa and smaller for more rigid gels (Trappmann et al., 2012)).

According to these assumptions we calculated the density of immobilized molecules. For the PA-gels used in cell cultures, 22 μL of the polymerization mix was added in between two glass coverslips, one silane-functionalized (15 mm diameter) and one non-functionalized (18 mm diameter). The gel formed uniformly between the glass coverslips but also formed on the extended surface of the larger 18 mm diameter coverslip. From the measured (by confocal microscopy) gel thickness of the re-hydrated PA-gels (25-50 μm, variations measured for different gel preparations) and the diameter of the slide (15 mm), we calculated the volume of the hydrated gels: 4.4-8.8 μL. We assumed that the hydrated PA-gels have a ~40% bigger volume than their initial polymerization mix (Hynd et al., 2007), thus the initial volume of the polymerization mix between the coverslips was ranging from 3 to 6 μL. Assuming that all of the streptavidin-acrylamide in the initial polymerization mix polymerized, we calculated the amount of molecules of streptavidin-acrylamide in this initial volume (Table A for reviewers). To determine the surface density of streptavidin-acrylamide, we assumed that biotinylated Abs and ICAM-1/Fc molecules could only access the streptavidin in the first 10 nm of the gels, which corresponds to the approximate diameter of a streptavidin molecule (Neish et al., 2002) and also, all of these molecules would be available for binding with soluble biotinylated antibodies taking into account the gel pore size (~15 nm). We thus found an effective surface density within a range of 20 to 40 streptavidin-acrylamide molecules/μm^2^. Since in most of the experiments a molar ratio of 10 anti-CD28+ICAM1-Fc to 1 anti-CD3 was used, the theoretical density of anti-CD3 is of 2 to 4 molecules/μm^2^ for 20 to 40 anti-CD28+ICAM1-Fc molecules/μm^2^ for the 0.5 kPa gels. Since the overall protein coating of PA-gels, measured by microscopy, was similar on the gels of different rigidities (Figure 1—figure supplement 1), this range of values for surface density applies to all PA-gels used in the study.

Table A

Hydrated Gel Thickness (μm)2550Hydrated Gel Volume (μL)4.48.8Volume of polymerization mix3.16.3Streptavidin-acrylamide moles in polymerized volume1.5E-112.9E-11Streptavidin-acrylamide concentration in the polymerized volume (μM)3.46.8Streptavidin-acrylamide moles in top 10 nm6E-151.2E-14Streptavidin-acrylamide molecules in top 10 nm3.6E+097.2E+09Effective Surface Density (molecules/μm^2^)20.340.7

As stated by the reviewers, it would be helpful to compare this density with the densities used in other studies. Alas, none of the published reports studying T cell sensitivity to stiffness provided the density of molecules bound to the substrate used for T cell activation (Judokusumo et al., 2012, O'Connor et al., 2012, Tabdanov et al., 2015, Bashour et al., 2014, Hu et al., 2016, Lambert et al., 2017). However, we can compare the densities used herein with the lowest densities of TCR ligands reported to induce T cell activation and immune synapse formation on artificial lipid bi-layers, i.e. 0.2 molecules/μm^2^ (Grakoui et al., 1999). We can also state that this density corresponds roughly to 1000 MHC-peptide complexes per APC (Harding and Unanue, 1990). The estimation of the activating molecule densities and the comparison with the literature is now stated in the new version of the manuscript.

*2) The speeds for T cell motility on the hydrogels are very fast and the reduction in speed on addition of anti-CD3 is modest – no prior studies would define this as a stop signal. It’s not clear why these values are so high as the frame rate is pretty typical at one frame every 5 seconds. The authors may want to check that the distances or times are calibrated correctly. Even in vivo arrest or stopping would have a threshold of <2 µm/min? Could there be tracking errors that are accounting for centroid displacement every 5 seconds that might lead to an artificial impression that the cells are moving rapidly? How is the cell centroid tracked? The morphologies shown in Figure 4 don't appear consistent with the fast motility values reported for the cells on the stimulatory hydrogels. The cells don't appear sufficiently polarized to move this fast.*

Again we want to thank the reviewers for their comments. Indeed, looking more critically at our data we noticed that for the velocity we took into account all the cells, including the ones that were not actually migrating but “jingling around” without really displacing on the substrate. This increased artificially the mean instantaneous velocity of the whole cell population. To avoid this problem, we followed the tracks of individual cells and obtained their mean instantaneous velocities and maximum distance travelled on the gels. We considered cells as either arrested, if their maximum displacement, i.e. the maximum distance travelled, was lower than 10 μm in the monitored 5 min, or migrating. Mean instantaneous velocities are now given only for migrating T cells. We counted the number of arrested T lymphoblasts and added this parameter to the results in the new figure that shows the percentage of arrested T lymphoblasts on the anti-CD3+aCD28+ICAM-1 coated PA-gels for 2 donors (Figure 1). As stated now in the new version of the Results, more T cells arrested on the stiffest (100 kPa) substrate than on the softer ones (0.5 and 6.4 kPa). Yet, the percentage of arrested cells never attained the value found on anti-CD3+aCD28+ICAM-1 coated glass slides, even when tested after 7h of culture of T lymphoblasts on the different PA-gels. These results thus suggest that at low density of TCR ligands (condition of this study) and in the range of physiological APC rigidities (Bufi et al., 2015), most of the T lymphocytes do not stop to form a synapse but rather form kinapses (Dustin, 2008). Yet, in more extreme conditions of rigidities (extracellular matrix, pathological tissues) more cells stop.

Concerning the scanning electron microscopy images shown (previously Figure 4, currently Figure 1), the reviewers are right. Because of the multiple treatments of the slides and washing conditions, we probably imaged mostly T lymphoblasts interacting strongly with the substrates (arrested cells). Indeed, a handful of images of migrating cells were observed (see Figure 8). This explains why the images are not consistent with the fast motility values reported. We have now clearly stated this bias in the new version of the manuscript.

Author response image 1.Scanning electron microscopy picture of a T cell with migrating morphology on a 0.5 kPa PA-gel coated with aCD3+aCD28+ICAM-1.**DOI:**
http://dx.doi.org/10.7554/eLife.23190.028

*3) Data showing T cell cytokine production on APC monolayer (subsection “T cell activation is potentiated by APC mechanical properties”) are striking. However, the stiffness of PDMS used in this section was not comparable to the Young's modulus of PA gels used throughout the studies. PDMS substrates with higher stiffness for cell study can certainly be achieved (See example: Palchesko et al., PLOS One, 2012, doi: 10.1371/journal.pone.0051499). Although the cytokine levels are different in the sample tested. The authors interpreted CD25 expression on APC is insensitive to stiffness in their tested conditions, but whether this expression is "graded" or "all-or-none" in response to the stiffness remains untested. It would be helpful if the authors include PDMS gels with higher stiffness such that at least three data points could be tested to provide a relationship. This would also be of interest in relation to O'Connor et al., 2013 (cited) where antibodies coated on stiffer PDMS hydrogels were less stimulatory than antibodies on a softer (50-100 kPa) substrate. It would be interesting if this optimum at intermediate PDMS stiffness is also detected here when an APC is inserted between the T cell and PDMS substrate.*

*Ideally, measurement of the local stiffness of the HeLa cells on the PDMS should be provided. The assumption is that the greater spreading of the cells on the stiffer gel means the cells are stiffer, but as other factors could impact the stiffness of the surface this needs to be measured.*

Again we would like to stress that the reviewers’ points were particularly useful to better characterize the co-culture model we used in our study. However, the first thing we would like to mention is that we initially tried to culture HeLa cells on PA-gels but were unable to succeed (and thus used commercial PDMS culture gels only available at 1.5 and 28 kPa). Indeed, fibronectin coating of PA-gel surfaces was probably too low to allow adhesion of HeLa cells that cannot adhere directly to the uncharged polyacrylamide. Following the reviewers’ advice, we tried, without success, to develop PDMS substrates of higher Young moduli and of “good quality” (flat homogeneous surfaces) to compare with the commercial ones we are using. Therefore, in order to address the point mentioned by the reviewers and use one higher stiffness value, we cultured the HeLa cells directly on fibronectin coated glass slides that have a Young modulus in the order of GPa (see below).

To gain insight into the mechanical properties of HeLa cells plated on substrates of different rigidities we first measured their area and showed that it increased between 1.5 and 28 kPa but did not increase between 28 kPa and GPa (glass) (Figure 7, Figure 7—figure supplement 1). These results suggested that the area and, subsequently, the stiffness of HeLa cells reached saturating values, as was also reported for fibroblasts and mesenchymal stem cells, at a substrate rigidity of around 20 kPa (Solon et al., 2007, Tee et al., 2011). We then directly measured the stiffness of the adherent HeLa cells and detected a small (although not significant at 5% confidence) increase in the Young modulus of cells plated on the 28 kPa PDMS versus the 1.5 kPa gel (Figure 7). This weak modulation of HeLa cells to substrate stiffness, as compared to what was reported for other cells (Solon et al., 2007, Tee et al., 2011) might be due to the different cell type used. It can also be due to the fact that we looked into the stiffness of HeLa cells in confluent monolayers, which were used to ensure a more homogeneous substrate for T cells and minimize their contact with the PDMS substrate. Differences of rigidity between individual adherent and confluent cells were indeed reported for other cell types (Stroka and Aranda-Espinoza, 2011).

It is thus important to note that in our co-culture model, T lymphocytes were not submitted to the large variation of stiffness imposed by the PDMS (1.5 to 28 kPa) but rather to the small variations of viscoelastic properties of the HeLa-CIITA APCs (in the order of few hundreds of Pa), which reached a plateau. Yet, cytokine production by T lymphoblasts co-cultured with HeLa cells followed the variation of Young’s moduli of HeLa cells: it increased between 1.5 and 28 kPa (Figure 7) and reached a plateau between 28 kPa and glass similarly to their spread area (Figure 7—figure supplement 1), suggesting that TCR/CD3 triggering of T cells is highly sensitive to APC stiffness. Concerning the absence of increase in CD25 expression in lymphoblasts co-cultured with HeLa cells plated on PDMS of different rigidities (Figure 7—figure supplement 1), it is not surprising since T lymphoblasts are subjected to a small variation of HeLa cell stiffness (1.4 to 1.7 kPa). Indeed, CD25 expression remained unchanged for bigger variations of Young moduli (0.5 to 6.4 kPa) on the activating PA-gels.

The new data have been introduced and discussed in the new version of the manuscript.

*4) This study would benefit from better consideration of these time-dependent issues by conducting several of the included experiments at very early (minutes), intermediate (several hours) and late (~24 hours). It is well established that Ag-depended responses emerge over time with differential phases. For example stop signals develop acutely and are reversed over the time frame of tens of minutes to hours. Additionally, while metabolic changes can emerge over hours in a transcriptionally regulated fashion, they can also be altered by diverse (including biomechanical) signals in seconds.*

*The decision to do the metabolic measurements off the PA substrates weakens them. It’s well known that acid production by T cells is a real time response to antigen recognition (see work from McConnell in late 1990s). So the responses might have been different if they were measured on the respective substrates rather than measured in a plastic well seemingly with no TCR ligands. An earlier time point should probably be selected also.*

The kinetic issue is indeed an important one. We thus did our best to look at different functions at different time points. Concerning the migration capacity of T lymphoblasts, we measured within the first 30 min of contact or 7h after contact (both for durations of 5 min). We did not see any differences in the motility of T lymphoblasts or in the percentage of arrested cells between these two time points. Although, as stated by the reviewers “stop signals develop acutely and are reversed over the time frame of tens of minutes to hours.”in vivo (Mempel et al., 2004), we are not aware of such behaviors in vitro and did not observe any changes in the first 24h even on glass.

Regarding the metabolic measurements performed in the extracellular flux analyzer, i.e. off the PA-gels, it is indeed a very good point. To address it, we measured lactate production in the supernatants of activated T lymphoblasts at different time points. This production reflected the glycolytic capacity of cells in the different conditions of culture. We were unable to see any significant changes before 6h of co-culture. This was probably due to the low density of anti-CD3 on the gels. Yet, at later time points anti-CD3 induced increased lactate production in all the conditions used but increase of lactate production with rigidity was observed only between the 6.4 kPa and 100 kPa conditions and not between the 0.5 and 6.4 kPa conditions (Figure 5). To reinforce these results we also measured the phosphorylation of the ribosomal protein S6 in the different conditions by FACS analysis. In the presence of anti-CD3, the percentage of phospho-rpS6^+^ T cells started to increase from 6h after activation and stably increased during time (Figure 5, Figure 5—figure supplement 1). Percentage of phospho-rpS6^+^ T cells did not change when rigidity went from 0.5 to 6.4 kPa, but was significantly increased for of the 100 kPa gel (Figure 5, Figure 5—figure supplement 1), correlating with lactate production in the different conditions.

In order to complete these data, we also used the Seahorse technology to measure overall glycolytic capacity and oxygen consumption of activated T lymphocytes. Because with this test, we did not observe any clear differences at 24h (Figure 5—figure supplement 1), we performed the analyses also at 48h. At this time point, both parameters demonstrated a potentiation with stiffness for the whole range tested (0.5 to 6.4 kPa and 6.4 to 100 kPa) (Figure 5). This test was used rather than the lactate measurement, because at this time point cells started to divide and the test allowed to get a measurement for the same number of cells. At this late time point we could detect an increase (although small) of glycolytic capacity and maximal mitochondrial respiration between the 0.5 and 6.4 kPa and still observed an increase between the 6kPa and the 100 kPa conditions. Thus, as suggested by the reviewers, kinetics matters.

Together, these results show that at 24h variations in substrate stiffness within the range measured for APCs (hundreds to thousands of Pa) do not modulate metabolic changes in response to low density of TCR/CD3 stimulators (our conditions herein). In contrast, metabolic remodeling is increased by higher rigidities (100 kPa) observed for tissue, extracellular matrix or pathological environment. These results also show that response to rigidity builds on with time resulting in increased proliferation for the whole range of rigidity tested at 48h.

All these new data have been incorporated and are discussed in the new version of our manuscript.